# Cell softness renders cytotoxic T lymphocytes and T leukemic cells resistant to perforin-mediated killing

Yabo Zhou [1,9], Dianheng Wang[1,9], Li Zhou[1], Nannan Zhou[1], Zhenfeng Wang[1], Jie Chen[1], Ruiyang Pang[1], Haixia Fu[2], Qiusha Huang[2], Fang Dong[3], Hui Cheng [3], Huafeng Zhang[4], Ke Tang[5], Jingwei Ma[6], Jiadi Lv [1], Tao Cheng [3], Roland Fiskesund[7,8], Xiaohui Zhang [2] ✉ & Bo Huang [1,5] ✉

Mechanical force contributes to perforin pore formation at immune synapses, thus facilitating the cytotoxic T lymphocytes (CTL)-mediated killing of tumor cells in a unidirectional fashion. How such mechanical cues affect CTL evasion of perforin-mediated autolysis remains unclear. Here we show that activated CTLs use their softness to evade perforin-mediated autolysis, which, however, is shared by T leukemic cells to evade CTL killing. Downregulation of filamin A is identified to induce softness via ZAP70-mediated YAP Y357 phosphorylation and activation. Despite the requirements of YAP in both cell types for softness induction, CTLs are more resistant to YAP inhibitors than malignant T cells, potentially due to the higher expression of the drug-resistant transporter, MDR1, in CTLs. As a result, moderate inhibition of YAP stiffens malignant T cells but spares CTLs, thus allowing CTLs to cytolyze malignant cells without autolysis. Our findings thus hint a mechanical force-based immunotherapeutic strategy against T cell leukemia.

T cells possess superior ability to kill tumor cells[1]. This killing process occurs at immune synapses formed between CD8[+] cytolytic T cells (CTLs) and target tumor cells, where the T cells release pore-forming perforin and pro-apoptotic protease granzymes[2,3]. Perforin then creates pores in the tumor cell membrane, allowing the entry of granzymes into the cytoplasm to trigger tumor cell lysis[4,5]. Such pore formation on the plasma membrane represents the most proximal and essential step in the killing of tumor cells by CTLs[1,3,6]. Enigmatically,

perforin seems to act unidirectionally and does not form pores on the autologous membrane of T cells, thus allowing a single CTL to successively cytolyze multiple individual tumor cells[7].

Several hypotheses have been proposed to explain CTL resistance to autolysis[7–9], including the theories of lipid order and packing on lymphocyte membranes[10,11]. In addition, our recent studies revealed that mechanical softness (a parameter to describe deformation) prevents perforin pore formation in the target cell membrane[12]; in

[1]Department of Immunology & National Key Laboratory of Medical Molecular Biology, Institute of Basic Medical Sciences, Chinese Academy of Medical Sciences (CAMS) & Peking Union Medical College, Beijing, China. [2]Peking University People's Hospital, National Clinical Research Center for Hematologic Disease; Beijing Key Laboratory of Hematopoietic Stem Cell Transplantation, Peking University Institute of Hematology, Beijing, China. [3]State Key Laboratory of Experimental Hematology, National Clinical Research Center for Blood Diseases, Institute of Hematology & Blood Diseases Hospital, Chinese Academy of Medical Sciences & Peking Union Medical College, Tianjin, China. [4]Department of Pathology, School of Basic Medicine, Tongji Medical College, Huazhong University of Science and Technology, Wuhan, China. [5]Department of Biochemistry and Molecular Biology, Tongji Medical College, Huazhong University of Science and Technology, Wuhan, China. [6]Department of Immunology, School of Basic Medicine, Tongji Medical College, Huazhong University of Science and Technology, Wuhan, China. [7]Department of Clinical Immunology and Transfusion Medicine, Karolinska University Hospital, Stockholm, Sweden. [8]Department of Medicine, Karolinska Institutet, Huddinge, Sweden. [9]These authors contributed equally: Yabo Zhou, Dianheng Wang. ✉e-mail: zhangxh@bjmu.edu.cn; tjhuangbo@hotmail.com

contrast, mechanical stiffness (inverse to softness) is required for perforin to form pores in tumor cells[13–15], suggesting that mechanical parameters may be involved in CTL resistance.

These findings prompted us to hypothesize that effector CTLs use softness to prevent perforin pore formation. Given that many traits of T cells are also shared by malignant T cells[16–19], it is possible that malignant T cells are also resistant to perforin by using a mechanism similar to that of T cells, thus abetting leukemia progression. In support of this, malignant T cells, when interacting with CTLs, form impaired immunological synapses and thus have a worse prognosis than B cell malignancies[17,20,21]. Currently, it remains unclear whether malignant T cells exploit perforin resistance to evade CTL killing.

In this study, we provide evidence that effector CTLs exhibit the perforin-resistant softness by downregulating cytoskeletal protein filamin A (FLNA), which is used by leukemic T cells to evade T cell killing. Our findings provide new insights into the pathway by which CTLs evades autolysis at the immune synapse.

## Results

### Mechanical softness mediates CD8+ effector T cell resistance to perforin

Using OVA-specific CTLs derived from OT-I TCR transgenic mice and B16 melanoma tumor cells expressing the model tumor antigen ovalbumin (OVA), we conducted an in vitro cytolytic assay by incubating activated OT-I cells (effector) with OVA-B16 cells (target), and evaluated the killing efficiency using propidium iodide (PI) staining. We found that PI selectively entered target cells but not effector cells in a perforin-dependent manner, shown by perforin knockout blocking the entry of PI into target cells (Fig. 1a, b), consistent with previous reports[6,22]. Moreover, we found that PI effectively entered perforin-treated OVA-B16 cells, but rarely entered CD8+ T cells, even under 50 ng/ml high dose perforin conditions (Supplementary Fig. 1a). Examining allogeneic response in T cells showed that although CD8+ T cells from healthy donors effectively cytolyzed

human MCF-7 tumor cells, little PI entered CD8+ T cells (Supplementary Fig. 1b), suggesting CTLs are resistant to perforin-mediated autolysis. Perforin pore formation not only involves chemical factors but also requires mechanical stiffness[10,12,23], and cell softness abrogates this process, prompting us to speculate that CTLs exploit softness to interfere with perforin action and avoid autolysis. Using atomic force microscopy (AFM)[24,25], we demonstrated that the majority of both B16-OVA and MCF-7 tumor cells had a stiffness of >800 Pa; in contrast, effector T cells (human or mouse) had a stiffness of <400 Pa (Fig. 1c, and Supplementary Fig. 1c). Jasplakinolide (Jas) is a natural cyclodepsipeptide that can enhance cell stiffness by inducing actin polymerization[26]. We found that Jas pretreatment facilitated perforin pore formation in effector T cells (Fig. 1d). On the other hand, latrunculin A (Lat-A), an inhibitor of F-actin that can soften tumor cells[27], hindered the entry of PI into tumor cells (Supplementary Fig. 1d, e). Previous visualization of cell membrane pore formation was based on results from artificial liposomes using electron microscopy, showing pore sizes around 10–20 nm[28–30]. Differently, we developed a method to directly visualize membrane pores in tumor and immune cells using PeakForce Tapping mode AFM, showing much larger pore sizes[12,24,31]. Adapting this method, AFM showed that stiffening Teff cells allowed perforin pore formation; in contrast, softening naïve T cells impaired pore formation (Fig. 1e and Supplementary Fig. 1f). In addition, in line with these results, naïve T cells became softer upon activation (Fig. 1f). Together, these results suggest that CTLs use their intrinsic softness to protect against perforin forming pores.

### Filamin A downregulation induces CTL softness

To investigate the molecular basis by which CTLs become softer, we used a CRISPR/Cas9-based gene overexpression method[32,33], we screened membrane, or membrane-related, protein molecule(s) by transfecting Jurkat cells with a membrane protein gRNA library, followed by perforin treatment and PI staining. The top 10% among PI

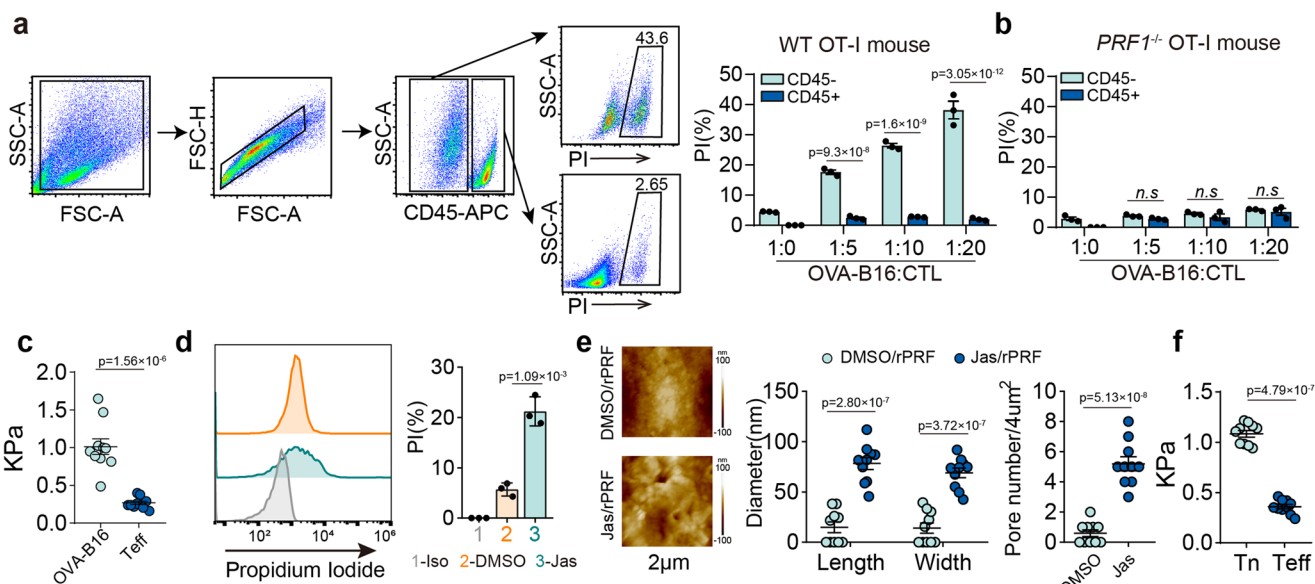

**Fig. 1 | Mechanical softness mediates the resistance of CD8+ effector T cells to perforin.** CD45− OVA-B16 cells were co-cultured with CD45+ OT-I CTLs (**a**) or *PRF1*−/− OT1-CTLs (**b**) at different ratios for 4 h. PI+ cells were analyzed by flow cytometry. **c** The stiffness of OVA-B16 or mouse CD8+ effector T (Teff) cells was determined by AFM. Each data point is the average of at least 20 force curve measurements of a single cell. **d, e** Human CD8+ effector T cells were pretreated with 200 nM Jas for 12 h. and then treated with Perforin for 10 min. The PI+ cells were analyzed by flow cytometry (**d**); Cells were fixed following 5 min treatment with 50 ng/ml Perforin

and imaged by AFM. The pores image was measured from 3 randomly selected areas on a single cell. The average size and number of pores formed were calculated within a cellular area of 2 × 2 μm (**e**). **f** The stiffness of mouse CD8+ naïve (Tn) and Teff cells was determined by AFM. PI Propidium Iodide, AFM atomic force microscopy. *n* = 3 independent experiments (**a**, **b** and **d**); *n* = 10 independent experiments (**c**, **e** and **f**). The data are represented as mean ± SD. *p* value by One-way ANOVA Bonferroni's test (**a**, **b**, **d** and **e**); *p* value by two-tailed Student's *t*-test (**c**, **e** and **f**). Source data are provided as a Source Data file.

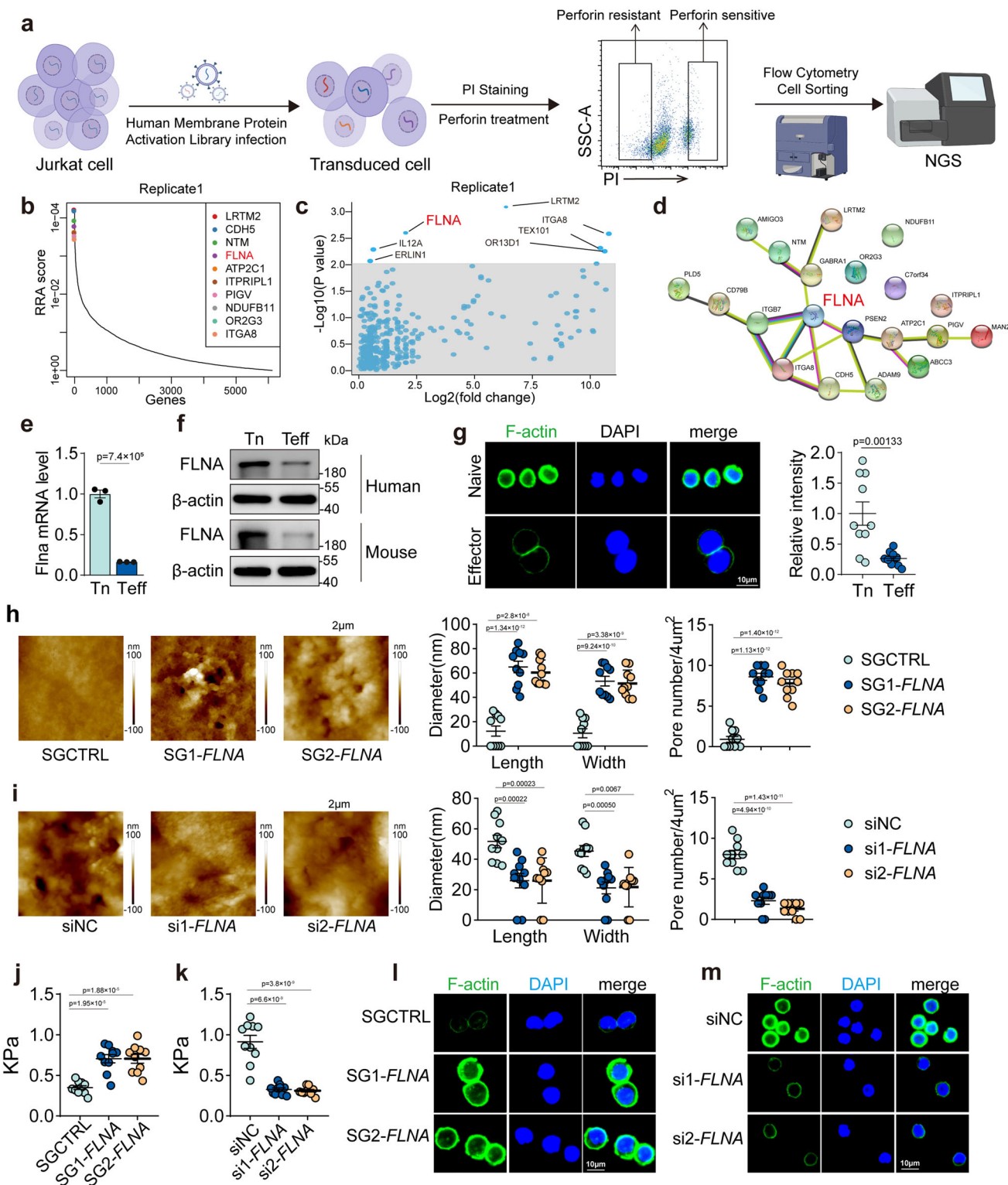

staining high (PI^h) and the bottom 10% among PI staining negative (PI^n) cells were sorted for next-generation sequencing (NGS) (Fig. 2a). Using the MAGeCK algorithm or RRA analysis[34], the top sgRNAs were identified in the PI^h cells (Fig. 2b). Among the top ten hits, filamin A (FLNA) caught our attention considering that FLNA is a hub molecule in protein-protein interaction networks, given its role as a cytoskeletal protein with the ability to crosslink actin filaments and thus stiffen cells (Fig. 2b–d). Two replicates further confirmed this result (Supplementary Fig. 2a–c). Both PCR and western blot showed that FLNA was downregulated in effector CTLs, compared to naïve

counterparts (Fig. 2e, f and supplementary Fig. 2d), with immuno-fluorescence further showing that effector CTLs had less F-actin staining than naïve CD8+ T cells (Fig. 2g). Overexpressing FLNA by CRISPRa, we found that perforin pore formation was enhanced in CTLs (Fig. 2h and Supplementary Fig. 2e, g); in contrast, knockdown of FLNA impeded the pore formation in perforin-treated naïve human CD8+ T cells (Fig. 2i and Supplementary Fig. 2f, h). In line with this, FLNA overexpression increased the stiffness of effector CTLs, and FLNA knockdown decreased the stiffness of naïve CD8+ T cells (Fig. 2j, k). In addition, the effector CD8+ T cells overexpressing FLNA

**Fig. 2 | Downregulation of filamin A is identified to induce CTL softness.**
**a** Schematic design of CRISPRa library screening in this study (Figure was created with BioRender.com). **b** Genes significantly enriched after Perforin treatment were identified through analysis of sequencing results in the MAGeCK program. **c** Significant hits from screens in cells treated with Perforin. Line indicates Bonferroni-corrected significance threshold. **d** PPI network based on the perforin sensitive associated hub genes of CRISPR screening. **e** The expression of *FLNA* in human CD8+ Tn and Teff cells were analyzed by real-time PCR. **f** The expression of FLNA in human (up) and mouse (bottom) CD8+ Tn and Teff cells were analyzed by western blot. **g** Human CD8+ Tn and Teff cells were stained with DAPI (blue), phalloidin (F-actin, green); Scale bar, 10 μm. **h** Human CD8+ Teff with or without *FLNA* overexpression by CRISPRa were treated with Perforin for 10 min and imaged by AFM. The formed pore size and number were calculated. **i** The same as (**h**), except that cells were human Tn cells with or without *FLNA* siRNA. **j** The stiffness of human CD8+ Teff with or without *FLNA* overexpression was determined by AFM. **k** The same as (**j**), except that cells were human Tn cells with or without *FLNA* siRNA. **l** Immunostaining of F-actin in human CD8+ Teff with or without *FLNA* overexpression. Scale bar, 10 μm. **m** Immunostaining of F-actin in human CD8+ Tn with or without *FLNA* siRNA. Scale bar, 10 μm. FLNA, Filamin A. *n* = 3 independent experiments (**e** and **f**); *n* = 10 independent experiments (**g**–**k**). The data are represented as mean ± SD. *p* value by One-way ANOVA Bonferroni's test (**h**–**k**); *p* value by two-tailed Student's *t*-test (**e**, **g**). Source data are provided as a Source Data file.

contained large amounts of F-actin, and *FLNA* knockdown reduced F-actin in naïve CD8+ T cells (Fig. 2l, m). These results suggest that effector CTLs use the downregulation of FLNA to avoid perforin pore formation and evade autolysis. However, when we conducted co-immunoprecipitation experiment, we did not detect the interaction between perforin and FLNA, suggesting that perforin does not directly interact with FLNA (Supplementary Fig. 2i).

## YAP mediates FLNA downregulation in activated CTLs

To determine the molecular mechanism by which FLNA is downregulated, we first looked at NFATc1, a key downstream transcription factor of TCR signaling, which is translocated from the cytosol into the nucleus upon dephosphorylation, transactivating a wide array of genes[35–37]. Blocking NFATc1 nuclear translocation with cyclosporine A (CsA) did not affect FLNA downregulation (Supplementary Fig. 3a). Moreover, ChIP-qPCR did not show binding of NFATc1 to the *FLNA* promoter (Supplementary Fig. 3b), suggesting that FLNA is regulated by another downstream molecule(s) of TCR signaling. YAP, a downstream transcription regulator of the Hippo signaling pathway, can also act as a mechanosensor[38]. Coincidentally, YAP has been reported to be activated by TCR signaling[39–41]. After 24 h of anti-CD3 and anti-CD28 stimulation, YAP was upregulated in CD8+ T cells, while knockdown of *YAP* by siRNAs resulted in FLNA upregulation (Fig. 3a, b and Supplementary Fig. 3c–e). A similar result was obtained using verteporfin, a YAP inhibitor (Fig. 3c and Supplementary Fig. 3f). In addition, using the proteasome inhibitor MG132 to treat cells, we found that FLNA expression was not altered in activated CD8+ T cells (Supplementary Fig. 3g), suggesting that FLNA downregulation is not mediated by proteasomal degradation. On the other hand, ChIP-qPCR showed that YAP did indeed bind to the *FLNA* promoter (Fig. 3d and Supplementary Fig. 3h). Using a dual-luciferase assay, we found that YAP did suppress *FLNA* promoter activity (Fig. 3e and Supplementary Fig. 3i). Meanwhile, the use of *YAP* siRNA or inhibitor could elevate the stiffness of activated human CD8+ T cells (Fig. 3f, g). In line with these results, more F-actin was observed in human CD8+ T cells transfected with *YAP* siRNA or treated with the inhibitor (Fig. 3h, i). Owing to the lack of a DNA-binding domain, YAP acts as a transcription co-activator to exert its function by mainly interacting with the transcription factor TEAD1[42]. We found that TEAD1 expression was upregulated in the activated CD8+ T cells (Fig. 3j and Supplementary Fig. 3j, k), and *TEAD1* knockdown elevated stiffness and relieved the inhibitory effect of YAP on FLNA in activated T cells (Fig. 3k, l). In line with these results, the *FLNA* promoter included a TEAD1 binding site (5′-CCTTA-3′), and the binding of TEAD1 to the promoter was verified by the ChIP−PCR assay (Fig. 3m and Supplementary Fig. 3l). Luciferase assay further verified the repression of TEAD1 on *FLNA* expression (Fig. 3n and Supplementary Fig. 3m). However, mutating the binding sequence to 5′- CCGTA −3′ disrupted the above effect of TEAD1 (Fig. 3n). Together, these results suggest that YAP mediates the downregulation of FLNA expression by binding to TEAD1 in activated CTLs.

## FLNA downregulation via YAP induces softness of T-leukemic cells

T-leukemic cells originate from the clonal proliferation of T cells, sharing many features with T cells[16–19]. We wondered whether FLNA downregulation was passed on to leukemic T cells. By analyzing gene expression profiles of leukemic T cells (*n* = 117) from the GEO database, we found, to our surprise, that *FLNA* was the most downregulated gene in leukemic T cells, when compared to normal bone marrow (BM) controls (*n* = 7) (Fig. 4a, b). Analysis of primary adult leukemic T cells showed that the cells expressed the lowest levels of *FLNA* among normal human naïve and activated CD8+ T cells and BM cells (Fig. 4c). Using a Notch (intracellular Notch 1, ICN1)-induced T-cell leukemia mouse model[43], we found that isolated murine T-leukemic cells also had lower FLNA levels than activated or non-activated normal splenic or thymic T cells (Fig. 4d). In line with these results, leukemic cells were softer than resting or activated T cells (Fig. 4e). In parallel, increased levels of nuclear YAP were observed in T leukemic cells, compared to those in resting or activated T cells (Supplementary Fig. 4a). Furthermore, knocking down or inhibiting YAP resulted in FLNA upregulation and increased the stiffness of leukemic T cells (Fig. 4f, g and Supplementary Fig. 4b, c). Moreover, we used *YAP*-KO primary human leukemic T cells to construct sub-cells with either the nuclear localization sequence (NLS)-*YAP* or nuclear export sequence (NES)-*YAP*. As expected, only NLS-*YAP*, but not NES-*YAP*, downregulated FLNA in primary T leukemic cells and thereby decreased cell stiffness (Fig. 4h, i). Similar results were obtained for Molt4 cells (Supplementary Fig. 4d–g). S127 dephosphorylation allows YAP to be present in an active state, promoting entry to the nucleus[44]. Surprisingly, we found that while more YAP was found in the nuclei of T-leukemic cells relative to activated CD8+ T cells, nuclear YAP was present in the Y357 phosphorylated form (Fig. 5a). This result prompted us to test LCK and ZAP70, two pivotal downstream molecules of TCR signaling which mediate tyrosine phosphorylation and are active in various tumor types[45–49]. LCK and ZAP70 displayed higher phosphorylation in primary human T-leukemic cells than in activated or resting CD8+ T cells (Fig. 5b). Transfection of 293 T cells with YAP-Flag and LCK-Myc or ZAP70-HA vector showed that YAP bound to LCK/ZAP70 (Fig. 5c). The in vitro kinase assay further showed that ZAP70 directly phosphorylated YAP; however, this phosphate group was removed by the addition of protein phosphatase (Fig. 5d). Meanwhile, use of LCK or ZAP70 inhibitors resulted in the disruption of YAP tyrosine phosphorylation (Fig. 5e and Supplementary Fig. 4h), concomitant with FLNA upregulation and increased cell stiffness (Fig. 5f and Supplementary Fig. 4i). Similar results were obtained from the knockdown of LCK or ZAP70 (Fig. 5g–i and Supplementary Fig. 4j–l). Additionally, the YAP mutant (alanine replacing tyrosine, Y375A) led to FLNA upregulation and increased cell stiffness (Fig. 5j, k). Together, these results suggest that FLNA downregulation induces T-leukemic cell softness via YAP Y375 phosphorylation.

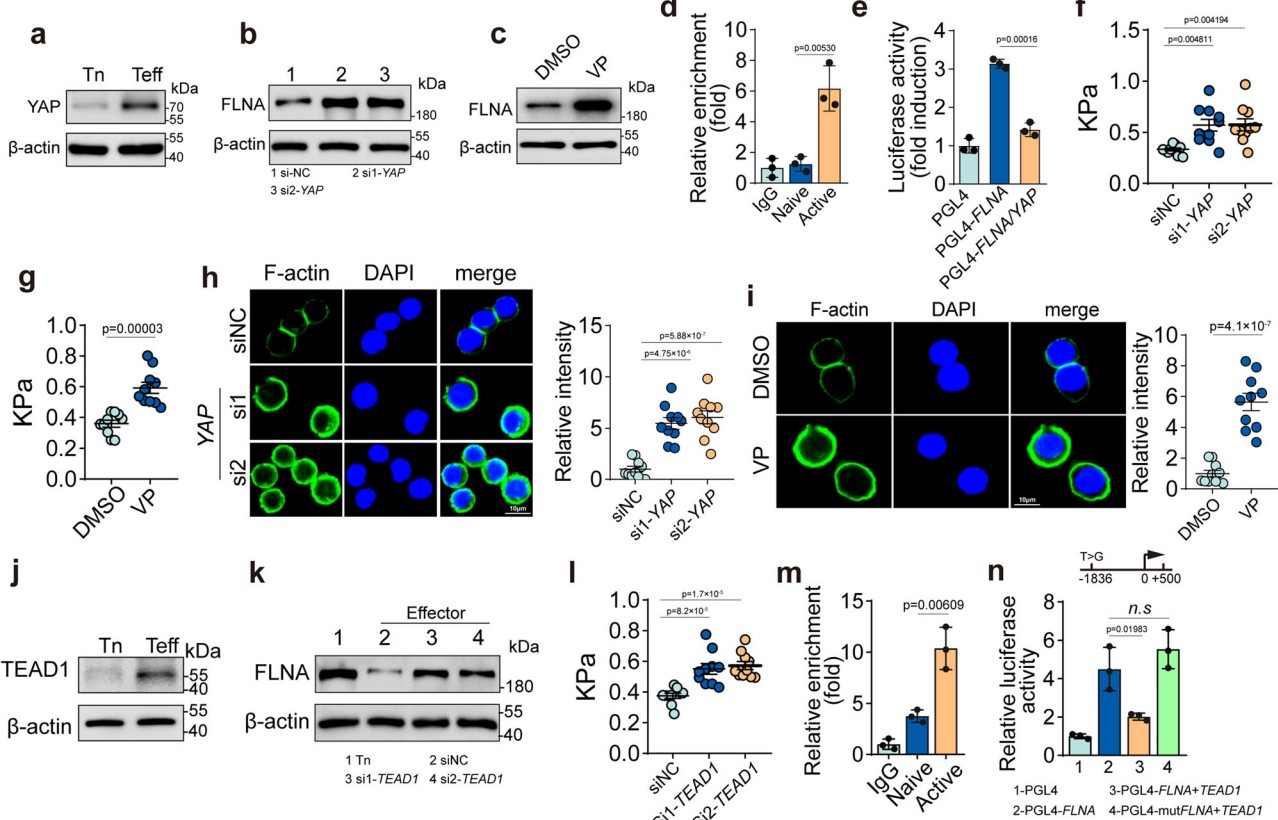

**Fig. 3 | YAP mediates FLNA downregulation in activated CTLs. a** The expression of YAP in human CD8+ Tn and Teff cells were analyzed by western blot. **b** Human CD8+ Teff cells were transfected with or without *YAP* siRNA, and FLNA expression was determined by western blot. **c** Human CD8+ Teff cells were treated with DMSO or verteporfin (VP) (1 μM) for 24 h, and FLNA expression was determined by western blot. **d** ChIP−qPCR analysis of *YAP* enrichment around the promoters of *FLNA* in human CD8+ Tn and Teff cells. **e** 293 T cells were co-transfected with *FLNA* promoter-luciferase reporter PGL4 and *YAP* plasmid for 24 h, followed by analysis of luciferase activity. **f** The stiffness of human CD8+ Teff cells transfected with or without *YAP* siRNA was determined by AFM. **g** The stiffness of human CD8+ Teff cells treated with DMSO or verteporfin (1 μM) for 24 h was determined by AFM. **h** Human CD8+ Teff cells transfected with or without *YAP* siRNA were stained with DAPI (blue), phalloidin (F-actin, green); Scale bar, 10 μm. **i** Human CD8+ Teff cells

treated with DMSO or verteporfin (1 μM) for 24 h were stained with DAPI (blue), phalloidin (F-actin, green); Scale bar, 10 μm. **j** The expression of TEAD1 in human CD8+ Tn and Teff cells were analyzed by western blot. **k** The expression of FLNA in human CD8+ Tn cells or Teff cells transfected with or without *TEAD1* siRNA was determined by western blot. **l** The stiffness of human CD8+ Teff cells transfected with or without *TEAD1* siRNA was determined by AFM. **m** ChIP−qPCR analysis of *TEAD1* enrichment around the promoters of *FLNA* in human CD8+ Tn and Teff cells. **n** 293 T cells were co-transfected with WT or mutant *FLNA* promoter-luciferase reporter PGL4 and *TEAD1* plasmid for 24 h, followed by analysis of luciferase activity. *n* = 3 independent experiments (**a**−**e**, **j**, **k**, **m** and **n**); *n* = 10 independent experiments (**f**−**i** and **l**). The data are represented as mean ± SD. *p* value by One-way ANOVA Bonferroni's test (**d**−**f**, **h** and **l**−**n**); *p* value by two-tailed Student's *t*-test (**g** and **i**). Source data are provided as a Source Data file.

## T-leukemic cells usurp the softness to evade CTL killing

Next, we investigated whether T-leukemic cells used softness to evade perforin-mediated CTL killing. Using perforin to treat primary T and B leukemic cells, we found that perforin treatment resulted in expansive pore formation in leukemic B cells and allowed abundant PI into the cells; however, few pores were formed in T-leukemic cells and rare PI was included in the cells (Supplementary Fig. 5a, b). By transfecting or treating primary T leukemic cells with FLNA-CRISPRa vector, *YAP* KO vector, YAP inhibitor or Jas, we found that perforin pore formation was enhanced, concomitant with increased PI entry (Supplementary Fig. 5c–j). To validate the results in vivo, we constructed OVA-expressing ICN1 leukemic cells, and adoptively transferred the OVA-ICN1 cells into NSG mice, followed by treatment with Jas, OT-I cells transfer or Jas/OT-I (Fig. 6a). Two days later, we found that Jas treatment, although increasing ICN cell stiffness, did not cause the entry of PI into the ICN1 cells, and the transfer of OT-I cells alone resulted in a very low level of PI level into the leukemic cells; however, combined treatment resulted in the entry of substantial PI into cells (Fig. 6b, c). Notably, using Jas to treat tumor cells in vitro for 24 h and removed the Jas by refreshing culture medium, 24 or 48 h later, we found that Jas-treated tumor cells were still stiffer than the untreated control cells

(Supplementary Fig. 6a). A consistent result was obtained from the removal of Lat A, an agent that softens cells, in Lat-treated YAP1-KO tumor cells (Supplementary Fig. 6b). In addition, the combination of Jas with gp100-specific or *PRF1−/−* OVA-specific CD8+ T cells did not generate the combined effect in vivo (Supplementary Fig. 6c, e). In line with these results, we found that OVA-ICN1-bearing NSG mice succumbed to T-cell leukemia and died around 3 weeks, which was not affected by Jas treatment; in contrast, Jas/OT-I treatment led to most mice being alive (Fig. 6d), which was much better than that of a single OT-I transfer. Moreover, we found that Jas/OT-I treatment strikingly reduced the accumulation of GFP+ ICN1 cells in the BM and spleen of mice (Fig. 6e, f). As expected, the efficacy of Jas/T cells was abrogated by the transfer of gp100-specific or *PRF1−/−* CD8+ T cells (Supplementary Fig. 6d, f). Apart from the ICN1 mouse model, we also constructed human CD1a-specific CAR T cells which can recognize T leukemic Molt4 cells[50]. Following the transfer of 2 × 10^6 *FLNA*-overexpressing or vector-control Molt4-luc cells into the NSG mice (Fig. 7a), tumor growth and cell cycle did not differ between the two groups (Fig. 7b, c and Supplementary Fig. 6g). Intriguingly, treatment of mice with CD1a-CAR T cells, which target T-leukemic cells, markedly suppressed *FLNA*-overexpressing Molt4-luc cell growth and prolonged the survival of

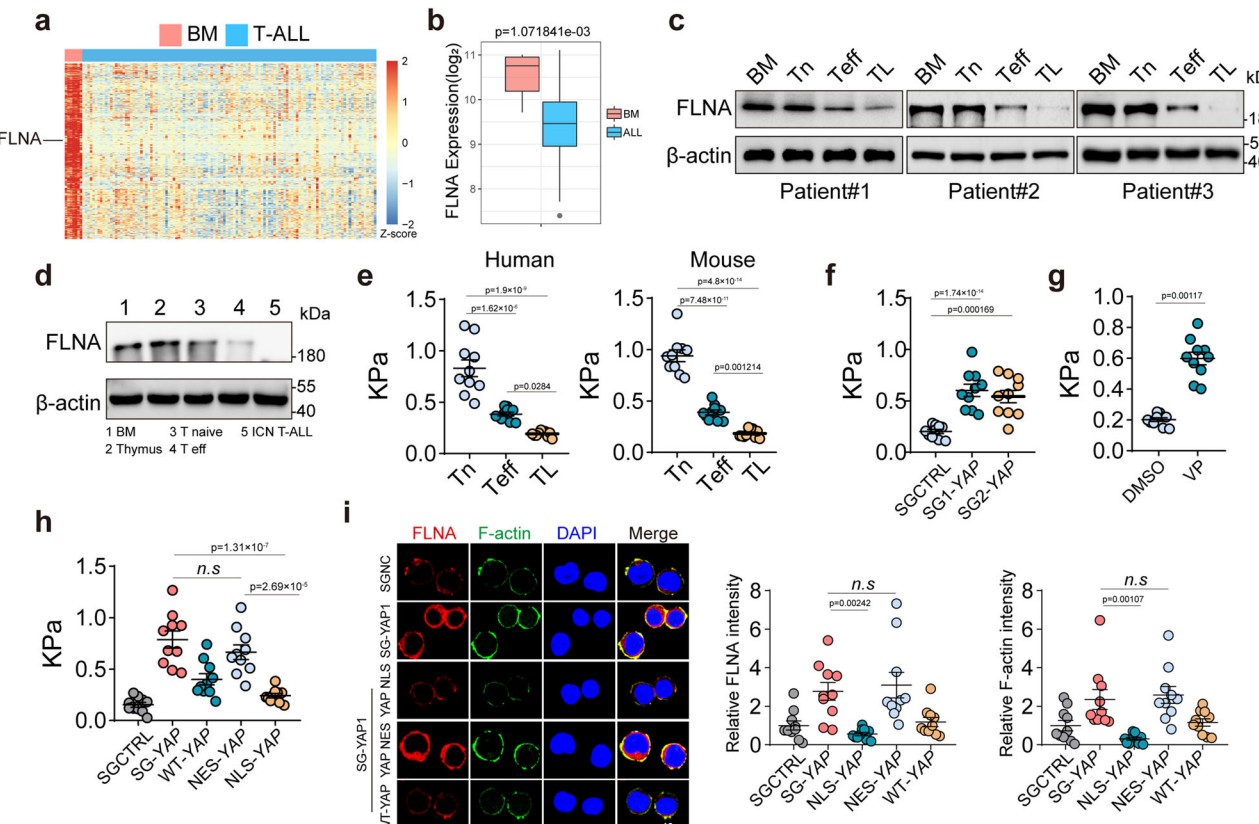

**Fig. 4 | FLNA downregulation via YAP induces the softness of T-leukemic cells.** **a** Heatmap of top 200 lowly expressed genes in T-ALL samples ($n = 117$) in comparison to 7 normal bone marrow samples (BM). **b** Distributions of *FLNA* mRNA expression derived from the heatmap. **c** The expression of FLNA in BM cells, naïve (Tn) and effector CD8+ T cells (Teff) and T leukemic cells (TL) were analyzed by western blot. **d** The expression of FLNA in BM, thymus, naïve and effector CD8+ T cells and T-leukemic cells from C57BL/6 mouse were analyzed by western blot. **e** The stiffness of human (left) or mouse (right) CD8+ Tn, Teff and TL was determined by AFM. **f** The stiffness of human primary T leukemic cells transfected with or without *YAP* sgRNA was determined by AFM. **g** The stiffness of human primary T

leukemic cells treated with DMSO or verteporfin (1 μM) for 24 h was determined by AFM. **h, i** The stiffness of SGCTRL, *YAP*-SG, *YAP*-SG/NLS-*YAP*, *YAP*-SG/NES-*YAP* or *YAP*-SG/WT-*YAP* T leukemic cells was analyzed by AFM (**h**). cells were stained with DAPI (blue), phalloidin (F-actin, green) and FLNA (red). Scale bar, 10 μm (**i**). T-ALL, T cell acute lymphoblastic leukemia. $n = 3$ biologically independent samples (**c**); $n = 3$ independent experiments (**d**); $n = 10$ independent experiments (**e–i**). The data are represented as mean ± SD. $p$ value by One-way ANOVA Bonferroni's test (**e**, **f**, **h** and **i**); $p$ value by two-tailed Student's *t*-test (**g**). Source data are provided as a Source Data file.

NSG mice (Fig. 7d–f). Together, these data suggest that T-leukemic cells use mechanical softness to impede perforin pore formation.

## CTL killing and autolysis are separated by low dose YAP inhibitor

Identifying the regulation of cellular softness by the YAP-FLNA axis provided a potential strategy to enhance CTL killing of T-leukemic cells via YAP inhibition. Using T-leukemia NSG mouse model, we found that CD1a CAR-T cells effectively destroyed *YAP1*-KO T leukemic cells, but not the control leukemic cells, leading to the prolonged survival of the mice (Fig. 8a–c); however, this killing process was abrogated by Lat-A treatment (Supplementary Fig. 7a–c). Similar results were obtained by transplanting *YAP*⁻/⁻ ICN cells into NSG mice (Supplementary Fig. 7d, e). We observed that such Lat-A treatment did not affect the effector function and stiffness of CAR-T cells (Supplementary Fig. 7f, g). These results suggest that YAP is a potential target to improve CTL killing of T cell leukemia, which, however, faced a dilemma in that YAP inhibition might increase CTL stiffness and, therefore, cause CTL autolysis. Activated T cells possess a remarkable ability to remove drug molecules[51], prompting us to hypothesize that low dose YAP inhibitor only increases the stiffness of T-leukemic cells but doesn't affect CTLs. We added different doses of the YAP inhibitor verteporfin (VP) to CTLs and T-leukemic cells. We found that high dose VP (1 uM) caused pore formation in both CTLs and target cells; however, low dose VP (0.1uM)

only resulted in the pore formation in target cells (Fig. 8d, e). Consistent with this, high performance liquid chromatography (HPLC) analysis showed that VP was mainly present in T-leukemic cells, but not in CTLs, concomitant with increased stiffness of leukemic cells (Fig. 8i, and Supplementary Fig. 8a). These results might be attributable to the expression of drug-resistant transporter MDR1 in CTLs[52]. We found that MDR1 was highly expressed in CD8+ effector T cells, but barely expressed in T-leukemic cells (Fig. 8f), and blocking MDR1 enhanced pore formation in CTLs by administering of low-dose VP (Fig. 8g, h). Similar results were obtained in primary human T-leukemic cells (Supplementary Fig. 8b), suggesting that a low-dose YAP inhibitor might enhance cytolysis while avoiding autolysis of CTLs. Given the lack of MDR1 in T-leukemic cells, we transfected them with pCMV-MDR1. We found that the transfection decreased Jas levels in Jas-treated T-leukemic cells, abrogated Jas-induced cell stiffness, and reduced perforin-mediated pore formation (Supplementary Fig. 8c–f). In contrast, use of MDR1 inhibitor increased Jas content, cell stiffness, and perforin-formed pores in Jas-treated CD8 + T cells (Supplementary Fig. 8g–j). In addition, we found that low-dose VP improved CTL activity by upregulating TNFα and IFN-γ (Fig. 8j), consistent with previous reports[40,41]. Then, we further clarified the in vivo treatment outcome by VP with a high dose (stiffening T cells and leukemic cells) and low dose (only stiffening leukemic cells) in the OVA-ICN1 mouse model (Fig. 8k, l). We found that a single high-dose, rather than low-

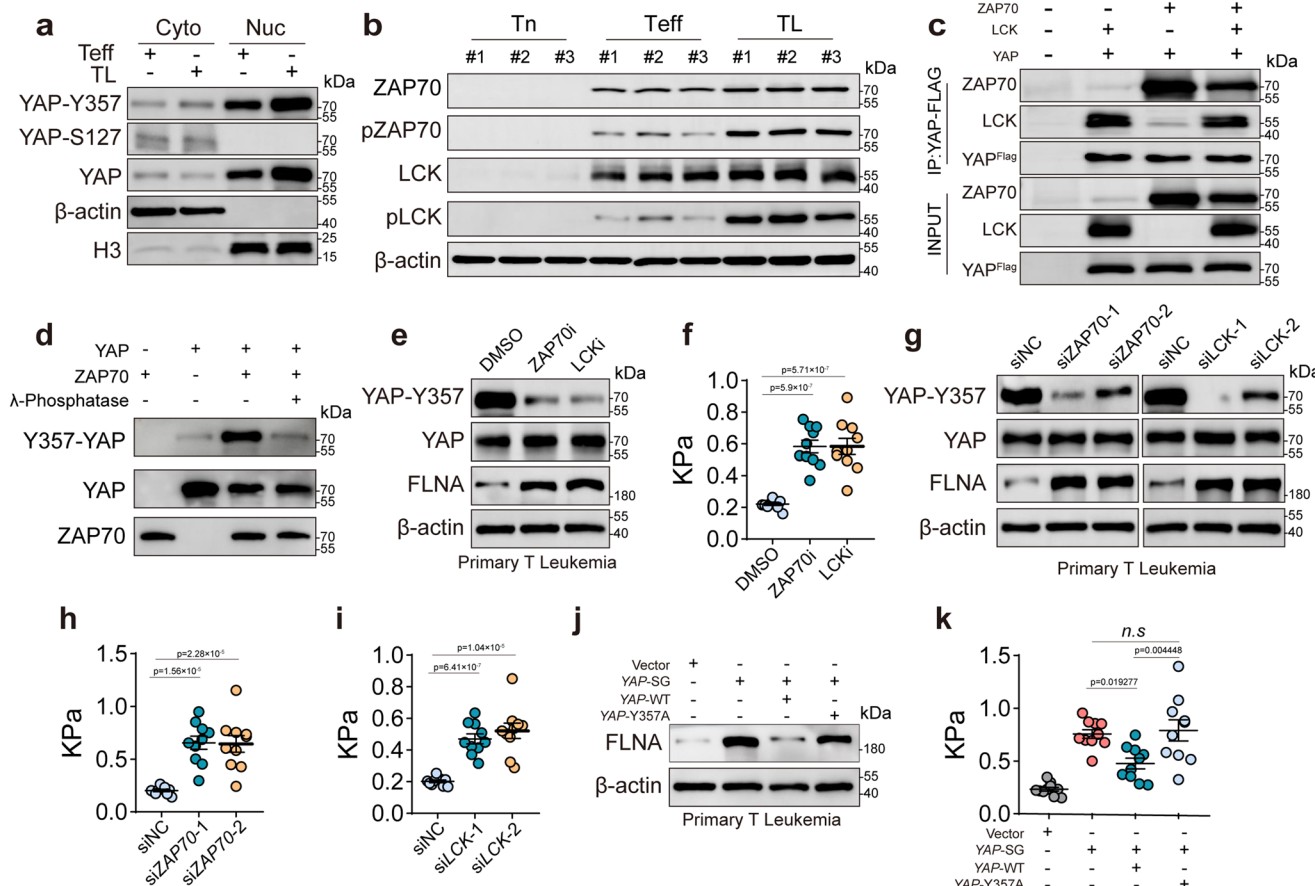

**Fig. 5 | Activation of TCR signaling phosphorylates YAP result in the softness of T-leukemic cells. a** The expression of cytosolic and nuclear YAP, S127-YAP and Y357-YAP in human CD8[+] Teff and T-leukemic cells was analyzed using western blot. **b** The expression of ZAP70, p- ZAP70, LCK and p-LCK in human CD8[+] Tn and Teff cells and TL cells was analyzed by western blot. **c** Immunoblot of immunoprecipitations of YAP from HEK293T cells transfected with the indicated combinations of ZAP70, LCK, and YAP expression vectors. the expression of ZAP70, LCK, and YAP were analyzed by western blot. **d** Purified recombinant human YAP1 was incubated with active ZAP70 for in vitro kinase assay. Phosphorylated proteins generated from the in vitro kinase reactions to treatment with or without Lambda Protein Phosphatase. Phosphorylated and total YAP1 and ZAP70 were determined by western blot. **e** The expression of FLNA, YAP and Y357-YAP in human primary T

leukemia treated with an inhibitor of ZAP70 or LCK were analyzed by western blot. **f** The stiffness of mouse CD8[+] Teff cells treated with an inhibitor of ZAP70 or LCK was determined by AFM. **g–i** The expression of FLNA, YAP and Y357-YAP of human primary T leukemia cells transfected by *ZAP70* or *LCK* siRNA were analyzed by western blot (**g**). The stiffness of cells transfected by *ZAP70* (**h**) or *LCK* (**i**) siRNA was determined by AFM. **j, k** The expression of FLNA in Vector, *YAP*-SG/WT-*YAP* and *YAP*-SG/*YAP*-Y357A human primary T leukemia cells was analyzed western blot (**g**), and the stiffness of cells was determined by AFM (**k**). *n* = 3 independent experiments (**a, c, d, e, g** and **j**); *n* = 10 independent experiments (**f, h, i** and **k**). The data are represented as mean ± SD. *p* value by One-way ANOVA Bonferroni's test (**f, h, i** and **k**). Source data are provided as a Source Data file.

dose, VP inhibited the OVA-ICN1 leukemia growth and decreased the number of tumor cells in the BM and spleen. However, when combined with OT-I cells rather than perforin knockout OT-I cells, high- and low-dose VP generated a similar inhibitory effect (Supplementary Fig. 8k, l). Intriguingly, the low-dose VP, however, resulted in better long-term survival of the mice, concomitant with intact liver and kidney functions, which were damaged in the high-dose group (Supplementary Fig. 8m–o). Together, these results suggest that low dose YAP inhibitor enhances CTLs to cytolyze T-leukemic cells without inducing autolysis in vivo.

**Softness promotes T-leukemic cell immune evasion in patients**
Finally, we explored a potential translation of the above findings to patients with adult T-cell leukemia, considering the strong activation of ZAP70 in malignant T cells of these patients[16]. We found that YAP was present in an active form, with Y357 phosphorylation, in primary adult T-leukemic cells isolated from patients (*n* = 10), concurrent with lower expression of FLNA relative to normal control T cells (Fig. 9a and Supplementary Fig. 9a). As expected, both FLNA and YAP Y357 phosphorylation levels were negatively and positively correlated with the

softness of T-leukemic cells (Fig. 9b–d). By co-culturing T-leukemic cells from patients and allogeneic CD8 + T cells isolated from healthy donors, we found that the T-leukemic cells were resistant to allogeneic killing relative to B-leukemic or myeloid leukemic cells (Fig. 9e). However, in FLNA-overexpressing or YAP Y357A-mutating T-leukemic cells, the killing resistance was disrupted (Fig. 9f, g). Moreover, when we used allogenic T cells to cytolyze the 10 primary leukemias, we found that the killing rate was positively and negatively correlated to FLNA and phosphorylated YAP (Y357), respectively (Supplementary Fig. 9a–c). Similarly, stiffening T-leukemic cells with Jas or VP also enhanced allogeneic killing (Fig. 9h, i). In addition, by analyzing the expression of HLA-A, B, and C, we found that HLA-A, B, and C were expressed in T-leukemic cells and were not affected by Jas, VP or FLNA overexpression (Supplementary Fig. 9d–f). To validate the in vitro results in vivo, we adoptively transferred isolated T-leukemic cells and allogeneic CD8[+] T cells into irradiated NSG mice (Fig. 9j). We found that the T-leukemic cells were minimally affected by allogeneic T cells (Fig. 9k, l), which, however, was reversed by YAP knockout. As expected, this YAP knockout-promoted cytotoxicity was abrogated by latrunculin A or FLNA/YAP dual knockout (Supplementary Fig. 9g, h).

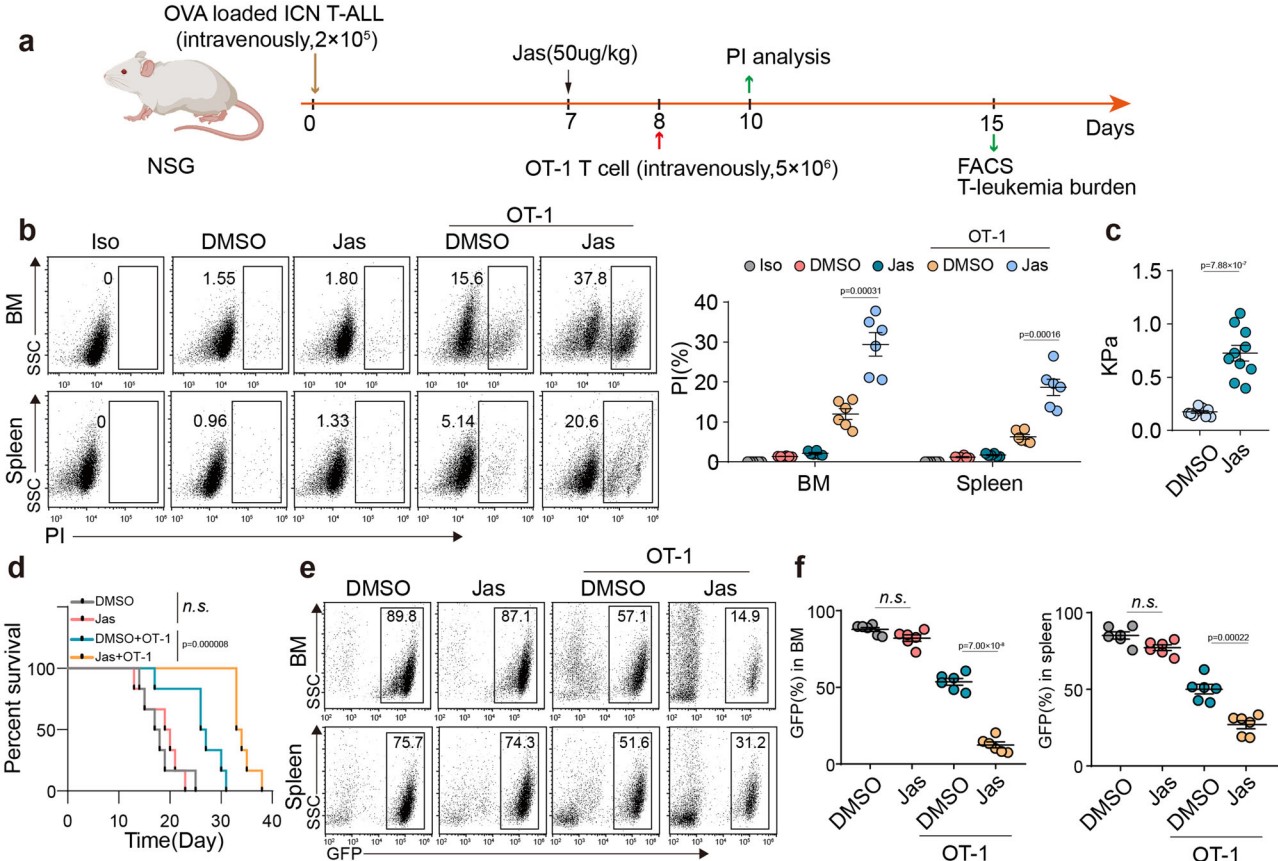

**Fig. 6 | T-leukemic cells usurp the softness to evade CTL killing in vivo.**
**a** Schematic of the mouse model is shown, NSG mice were transplanted with $2 \times 10^5$ OVA-expressing ICN1 leukemic cells. At 7 days post engraftment, NSG mice were treated or untreated with Jas (50 μg/kg), followed by adoptively transferring with or without $5 \times 10^6$ OT-I T cells. At 10 days post engraftment, PI⁺ cells in the ICN1 cells were analyzed by flow cytometry. At 15 days post engraftment, ICN1 leukemic cells dissemination were determined by flow cytometry (Figure was created with BioRender.com). **b** PI⁺ cells from spleen and bone marrow were analyzed by flow cytometry. Data from six individual mice were plotted and shown on the right.

**c** The stiffness of ICN treated with Jas in vivo was determined by AFM; $n = 10$ independent experiments. **d** Kaplan–Meier survival curves of OVA-expressing ICN1 leukemic cells in each group as indicated. ICN1 GFP⁺ cells from spleen and bone marrow were analyzed by flow cytometry (**e**); Data from six individual mice were plotted and shown on the right (**f**). ICN1, intracellular Notch 1. The data are represented as mean ± SD. *p* value by One-way ANOVA Bonferroni's test (**b** and **f**); *p* value by two-tailed Student's *t*-test (**c**); *P* value by Log-rank survival analysis (**d**). Source data are provided as a Source Data file.

In addition, the addition of Jas or low-dose YAP inhibitor markedly promoted allogeneic killing of T-leukemic cells (Fig. 9k, l and Supplementary Fig. 9i–k). Here, we found that Jas or Lat-A had no effect on the stiffness of effector T cells but did stiffen or soften T leukemic cells (Supplementary Fig. 10a, c). Consistently, the administration of low dose Jas or Lat-A did not affect the effector function of allogeneic T cells (Supplementary Fig. 10b, d). Together, these results suggest that T-leukemic cells in patients use their softness to attenuate CTL killing for immunoevasion.

## Discussion

Cells use both chemical and physical traits to regulate their behavior and responses. Previously, we demonstrated that highly tumorigenic tumor-repopulating cells use their mechanical softness to impede perforin pore formation, thus evading CTL immune attack[12]. In line with this, in the present study, we provide evidence that effector T cells also use softness as a weapon to evade perforin-mediated autolysis at the site of the immunological synapse. Moreover, we demonstrate that this T cell immuno-protective mechanism is shared by malignant T cells. Similar to activated CTLs, T-leukemic cells soften themselves by downregulating the cytoskeletal protein filamin A (FLNA). Thus, this study uncovers an immuno-escape pathway that is unique to T cell malignancies.

T cell-mediated cytolysis occurs at the site of the immunological synapse, where CTLs release granzymes and perforin, which drills pores in the target cell membrane, thus allowing the entry of granzymes and triggering cell apoptosis[1,3]. Paradoxically, the effect of perforin is unidirectional and does not form pores in the releasing cells[3,7,9,10]. Different mechanisms have been proposed to explain perforin-biased effects on target cell membranes. Surface cathepsin B in lytic granules is thought to degrade and inactivate perforin, thereby protecting CTLs from autolysis[9]. However, this hypothesis has not been proven in cathepsin B-deficient CTLs[53]. In addition, it has been proposed that glycosylated LAMP-1, with negative charges, prevents the binding of perforin to the CTL membrane, however this might be dispensable under some circumstances[10,54]. Previous studies have indicated that perforin binds to membrane phospholipid molecules in a Ca²⁺-dependent manner[8,23,55]. Recent reports further reinforce this notion by showing that lytic granules facilitate lipid membrane packaging to a higher-ordered level (lipid rafts), creating a negatively charged sink and inactivating perforin action on cytolytic T or NK cells[10,11]. However, the shortcoming of these studies is that they used liposomes instead of a true plasma membrane. Since the liposomal structure lacks protein components, the real effect of lipids in mediating perforin pore formation should be interpreted carefully[56]. Apart from these chemical effects, we believe that a physical effect is also

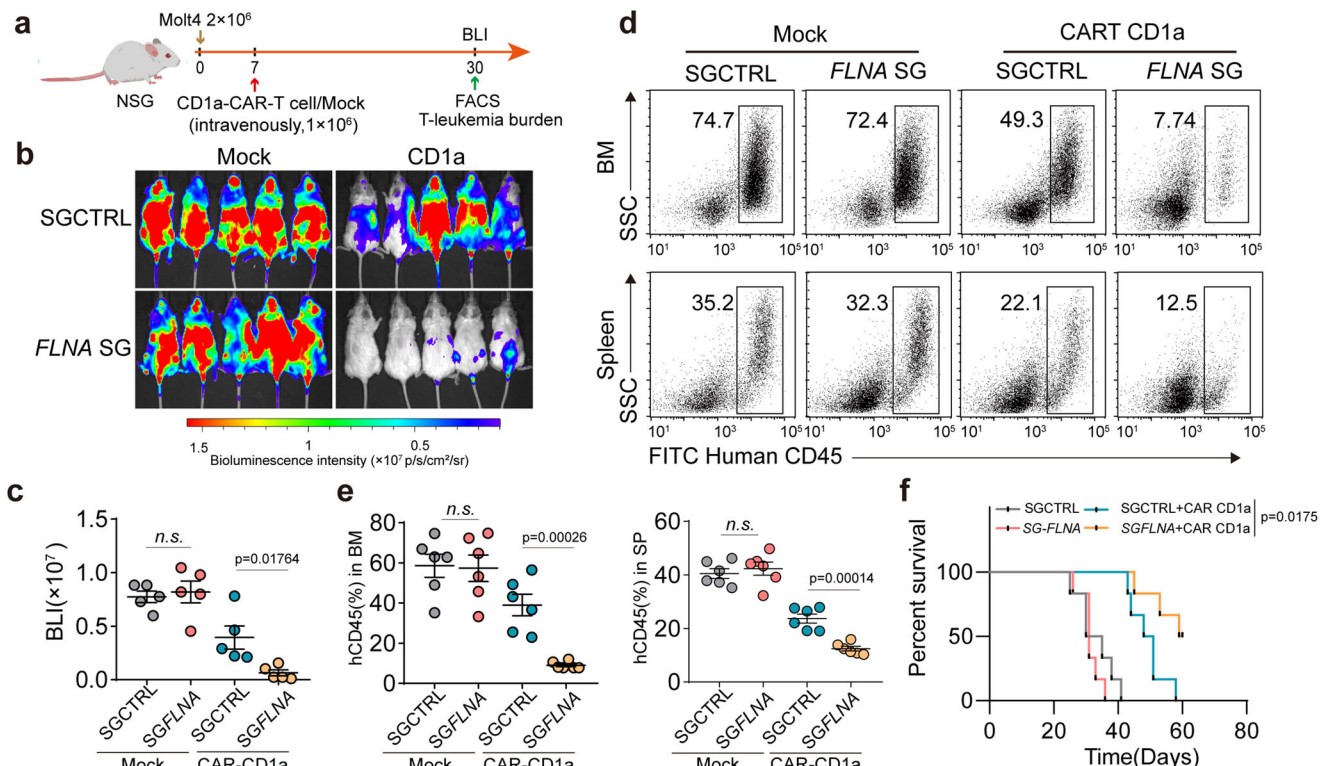

**Fig. 7 | FLNA improves CAR-T cell therapy efficacy in vivo. a** Schematic of the mouse model is shown, NSG mice were transplanted with $2 \times 10^6$ *FLNA*-over-expressing and vector control Molt4-luc cells. At 7 days post engraftment, NSG mice were adoptively transferred with $1 \times 10^6$ human CD1a-CAR or Mock T cells, then at 30 days post engraftment, 6 mice were taken out in each group and killed for assessment of leukemia burden (Figure was created with BioRender.com). **b** In vivo tumor bioluminescence imaging (BLI) of NSG mice treated with human CD1a-CAR or Mock T cells. **c** Leukemia burden was measured by quantification of BLI; $n = 5$ independent animals. **d**–**f** the same as (**b**) Human CD45$^+$ leukemia cells (CAR-T cells were excluded by gating strategy) from spleen and bone marrow were analyzed by flow cytometry (**d**); Data from six individual mice were plotted and shown (**e**). The data are represented as mean ± SD. $p$ value by One-way ANOVA Bonferroni's test (**c** and **e**); $p$ value by Log-rank survival analysis (**f**). Source data are provided as a Source Data file.

involved in the action of perforin. Our previous and current studies have both revealed that mechanical softness prevents perforin from forming pores in target tumor cells or autologous CTLs[12]. Based on findings from our group and others[12,14,57], we propose that activated T cells mobilize lytic granules and allow the release of perforin and granzymes into the immune synaptic space via membrane fusion, resulting in perforin pore formation in stiff target cells but not in soft autologous T cells.

Identifying T cell or T-leukemic cell resistance to perforin via FLNA downregulation is an important observation. As a V-shaped homo-dimer, FLNA can bind and cross-link cortical actin filaments into a three-dimensional structure through its N-terminal actin-binding domain, thus maintaining cell rigidity sensing[58,59]. FLNA down-regulation may therefore reduce cortical actin filaments, required for T cell resistance to perforin. Likely, CTL plasma membrane at the synaptic site is not completely covered by ordered domains (lipid rafts), and thus unordered membrane sections can be exposed for autologous perforin binding and pore generation. To avoid this, FLNA regulates perforin binding unordered membrane phospholipids, as FLNA-related actin filaments play an important role in lipid raft formation[60]. Here, we propose that high FLNA content transduces a strong force along actin filaments to perforin, thus stabilizing the conformation of perforin during its insertion into the target cell membrane for pore formation; however, FLNA downregulation redu-ces the magnitude of this mechanical force, thus impairing pore for-mation (Fig. 7l).

Yes-associated protein (YAP) is a crucial transcription activator that primarily forms a heterodimer with TEAD factors to bind target genes for transcription regulation[61]. YAP is appreciated in the context of the Hippo signaling pathway, where Mst1/2 phosphorylates and activates LATS1/2, which then phosphorylates at Serine 127 and inac-tivates YAP by inducing its cytoplasmic retention and degradation[44]. In this study, we found that FLNA downregulation is actually mediated by activated YAP-TEAD1. Although it has been known that TCR signaling can activate YAP, the underlying mechanism remains unclear[39–41]. The initial downstream molecules of TCR signaling are Lck and ZAP70, both of which are tyrosine kinases[62]. Intriguingly, ZAP70 is able to phosphorylate YAP at tyrosine 357, leading to YAP activation, con-sistent with a previous report[63]. This tyrosine-phosphorylated YAP is then transported into the nucleus, where FLNA is transcriptionally repressed[64]. Indeed, the use of verteporfin, a YAP inhibitor that dis-rupts YAP-TEAD interactions, increases FLNA expression levels and results in increased CTL stiffness. Therefore, during the interaction between effector T cells and target tumor cells at the synaptic site, activated TCR signaling leads to YAP activation and subsequent FLNA downregulation, thus triggering T cell resistance to perforin. Of note, ZAP70 activation seems to be a common feature of tumor cells, even for T-leukemic cells, transformed from very early progenitor T cells, which may lack TCR expression, thus guaranteeing YAP in an active form as well as the downregulation of FLNA in T-leukemic cells[47–49,65].

Our findings have implications for the development of anti-T leukemic therapeutics. YAP expression is deregulated in a variety of cancer types and exerts pro-tumor effects in a trimeric form by binding to its paralogous factor TAZ and a transcription factor such as TEAD, RUNX, or P73[66–69]. The oncogenic roles of YAP signaling are reflected by its sustained proliferation, inhibition of apoptosis, maintenance of

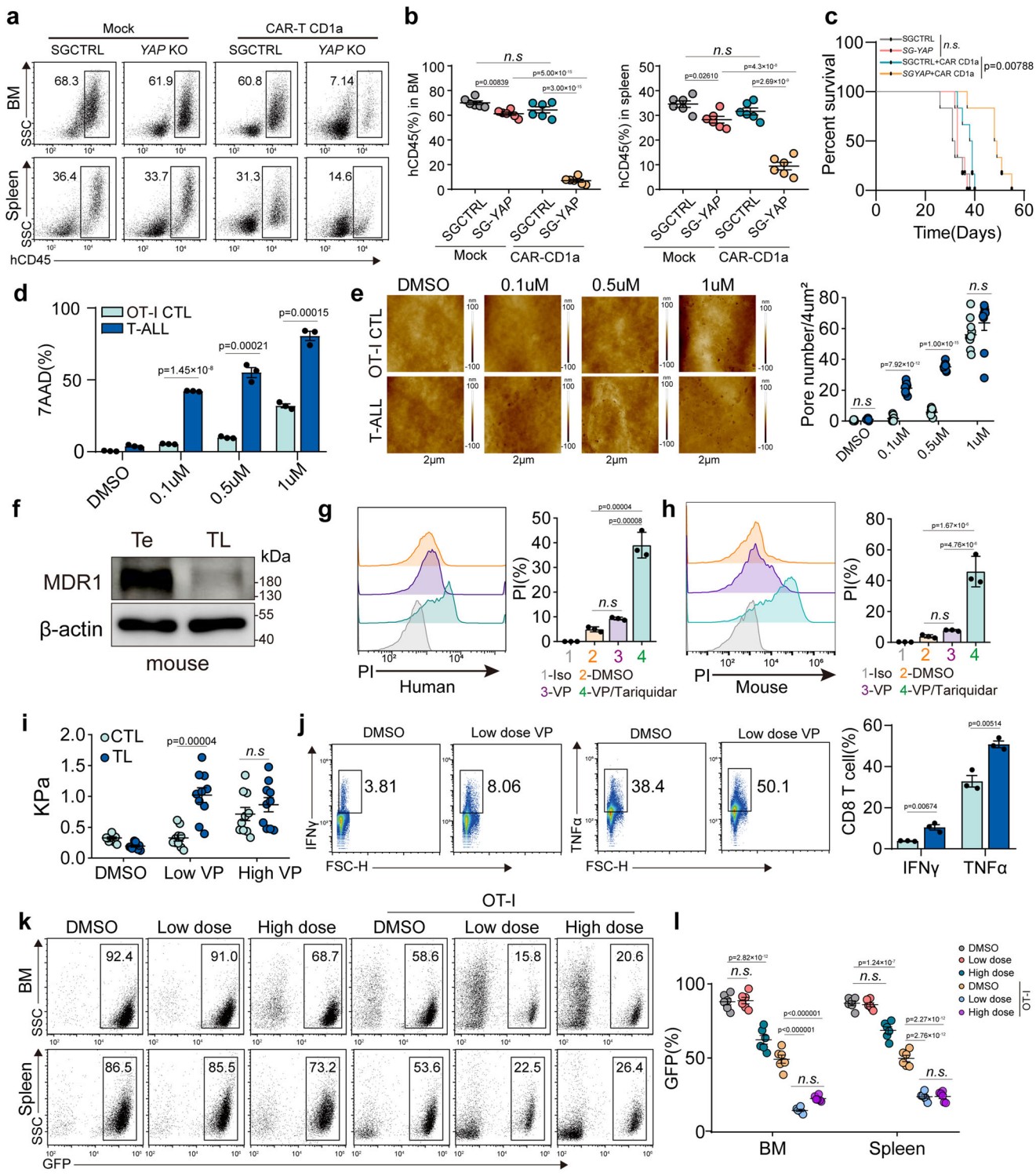

stemness, suppression of antitumor immune response, response to mechanical force, and development of treatment resistance[61]. Therefore, targeting YAP is an attractive therapeutic intervention for cancer treatment. In this study, we found that YAP is also highly expressed in the T-leukemic cells of patients. Notably, YAP activity promotes malignant cell growth but suppresses normal T cell function, thus providing a 'two birds, one stone' strategy that targets T-leukemic cells while simultaneously enhancing T cell killing. However, as However, as YAP is physiologically essential, complete inhibition of YAP activity might be too toxic to be used in clinical applications. Similar concerns are derived from the use of cytoskeletal drugs, which stably bind to F-

actin, thus resulting in cytotoxicity[70]. In this study, we found that once injection of Jas may maintain a 48-h stiffness of T leukemic cells, which might be explained by a stable binding of Jas to F-actin or that Jas-caused damage makes T leukemic cells difficult to recover the normal F-actin homeostasis within a short time. Despite these, in this study, we demonstrated that low dose YAP inhibitor is an ideal choice. We found that activated T cells and T-leukemic cells have different sensitivities to YAP inhibitors. CTLs highly express MDR1, which pumps drug molecules from the cytosol to the extracellular space[52]. Thus, in response to low-dose YAP inhibitor, CTLs can maintain an effective YAP activity, which transcriptionally suppresses FLNA expression, thus avoiding

**Fig. 8 | Moderate YAP inhibition improves CTL killing of T-leukemic cells but avoids autolysis. a, b** NSG mice were transplanted with $3 \times 10^6$ *YAP* knockout and vector control Molt4-luc cells. At 7 days post engraftment, NSG mice were adoptively transferred with $1 \times 10^6$ human CD1a-CAR-T cells. then at 30 days post engraftment, leukemia burden (spleen and bone marrow) was analyzed by flow cytometry (**a**); Data from six individual mice were plotted and shown (**b**). **c** the same as (**a, b**), the survival was analyzed. **d, e** OVA-expressing ICN1 leukemic cells were co-cultured with OT-I T cells treated with VP (0.1 uM, 0.5 uM, 1 uM) for 12 h. 7-AAD+ cells were analyzed by flow cytometry (**d**) or AFM (**e**). The formed pore size and number were calculated (**e**). **f** The expression of MDR1 in mouse CD8+ Teff cells and TL cells were analyzed by western blot. Human (**g**) and mouse (**h**) CD8+ T cells were pretreated with 0.1 uM VP combined with or without 1 nM Tariquidar for 12 h then treated with Perforin for 10 min. The PI+ cells were analyzed by flow cytometry. **i** The stiffness of ICN1 cells and CTLs treated with VP (0.1 uM, 1 uM) 12 h was determined by AFM. **j** Activated CD8 + T cells were stimulated with PMA and treated with 0.1 μM VP for 24 h. Flow cytometric analysis was performed to evaluate the expression of IFN-γ and TNFα. **k, l** NSG mice were transplanted with $2 \times 10^5$ OVA-expressing ICN1 leukemic cells. At 7 days post engraftment, NSG mice were adoptively transferred with or without $5 \times 10^6$ OT-I T cells, followed by with DMSO or VP (20 mg/kg as low dose or 200 mg/kg as high dose), into each allocated group every two days for 8 days. At 20 days post engraftment, leukemia cell dissemination was determined by flow cytometry (**k**), Data from six individual mice were plotted and shown (**l**). 7-AAD, 7-Aminoactinomycin D; PMA, Phorbol 12-myristate 13-acetate. $n = 3$ independent experiments (**d, f, h** and **j**); $n = 10$ independent experiments (**e** and **i**). The data are represented as mean ± SD. *p* value by One-way ANOVA Bonferroni's test (**b, d, e, g–j** and **l**); *p* value by Log-rank survival analysis (**c**). Source data are provided as a Source Data file.

autolysis by perforin; in contrast, such a YAP inhibitor can relieve the suppression of YAP on FLNA and enhance perforin-mediated cytolysis. Although we prove this concept in T-leukemic cells, the application of low-dose YAP inhibitor in other types of leukemia, lymphoma, or even solid tumors is worthy of further investigation. In addition, our findings may be useful to modify the current clinical testing of YAP-TEAD inhibitors, considering YAP-TEAD inhibitors are currently being tested in Phase I/ II clinical trials[71,72].

## Methods

### Animals and cell lines

Mouse tumor cell lines B16, OVA-B16 (melanoma) and human tumor cell lines MCF7 (breast cancer) were purchased from China Center for Type Culture Collection (Beijing, China) and grown in complete RPMI-1640 (Thermo Scientific, USA) with 10% fetal bovine serum (FBS; Gibco, USA). The 293 T, Jurkat, and MOLT4 cells were purchased from American Type Culture Collection (ATCC) and cultured in Advanced RPMI 1640 (Thermo Scientific, USA) with 10% fetal bovine serum (FBS; Gibco, USA), 1% penicillin/streptomycin (Gibco, USA), 1% non-essential amino acids (Gibco, USA), 2 mM L-glutamine (Gibco, USA), and 1 mM sodium pyruvate (Gibco, USA). The 293 T cell was maintained in Dulbecco's modified Eagle's medium (DMEM, Gibco, USA) containing 10% FBS and 1% penicillin/streptomycin (Gibco, USA). All cell lines were tested for Mycoplasma detection, interspecies cross contamination and authenticated by isoenzyme and short-tandem repeat analyses in Cell Resource Center of Peking Union Medical College before the study. Cell lines used in the experiments were within 20 passages.

All protocols and animal studies were performed in accordance with the Guide for the Care and Use of Laboratory Animals by the US National Institutes of Health. In addition, the Animal Care and Use Committee of Chinese Academy of Medical Science approved this study (ACUC-A02-2022-085). All mice, under a specific pathogen-free environment, were housed in a temperature-controlled environment of 22–23 °C, and 40–70% humidity in individually ventilated cages with wood pieces as bedding with 12 h light/dark cycles and received food and water ad libitum (Jiangsu Xietong Pharmaceutical Bio-engineering, cat.1010001, Jiangsu). For all in vivo experiments, mice allocated to different experimental groups were sex-, age- and housing-matched. Each group of mice were cohoused separately.

Six- to Eight-week-old male or female C57BL/6 J Nifdc (stock number: 000664) or NOD-SCID-Il2rg−/−(NSG) mice (stock number: 005557) were purchased from the Center of Medical Experimental Animals of Chinese Academy of Medical Science (Beijing, China). Pmel-1 transgenic (B6.Cg-Thy1a/Cy Tg(TcraTcrb)8Rest/J) mice were presented by Dr. Ying Wan (Third Military Medical University, China). *PRF1*−/− (C57BL/6-Prf1tm1Sdz/J, referred to as *PRF1*−/−,) mice were obtained from Shanghai Model Organisms Center, Inc (Shanghai, China). OT-I transgenic (C57BL/6-Tg (TcraTcrb)1100Mjb/J) mice were gifted by Dr. Hui Zhang (Sun Yat-Sen University, China) and were crossed to *PRF1*−/− mice. ICN mouse model was gifted by Dr. Tao Cheng (State Key Laboratory of Experimental Hematology, Institute of Hematology and Blood Diseases Hospital). All studies involving mice were approved by the Animal Care and Use Committee of Chinese Academy of Medical Science (ACUC-A02-2022-085).

Mice were checked daily for any signs of illness. Upon visible or palpable leukemia development or upon moribund appearance, mice were euthanized with a combination of $CO_2$ and cervical dislocation to guarantee the death of the animals. After that, the tissues were collected for further analysis.

### Human samples

The Human primary T-leukemic specimens were obtained from patients at the Peking University People's Hospital. All donors provided informed written consent before sampling according to the Declaration of Helsinki, and the present study was approved by the institutional ethics committees of Committee of Peking University People's Hospital (NKRDP2021005-EC-2). The clinical features of the patients are listed in Supplementary Table 1.

### RNA extraction and quantitative real-time PCR

Total cellular RNA was extracted using TRIzol (Invitrogen, USA) and random primed RNAs (1 μg) were reverse transcribed with RevertAid first-strand complementary DNA synthesis kit according to the manufacturer's instructions (Thermo Scientific, USA). Relative expression of the mRNA was calculated by $2^{-\Delta\Delta Ct}$ method and normalized to β-ACTIN. Real-time PCR was performed using ABI QuantStudio (Applied Biosystems, CA, USA). Values are means ± SD from three independent experiments which were performed in duplicate. Statistical comparisons among groups were performed using a One-way ANOVA followed by Boferroni's test. Specific PCR primer sequences are listed in Supplementary Table 2.

### Western blot analysis

Cells were collected, lysed in RIPA lysis buffer and sonicated. The protein concentrations were determined by a BCA kit (Beyotime, China). Then, the protein was run on an SDS-PAGE gel and transferred to nitrocellulose. Nitrocellulose membranes were blocked in 5% bovine serum albumin (BSA) and probed with antibodies overnight. Secondary antibodies conjugated to horseradish peroxidase were followed by enhanced chemiluminescence (Thermo Fisher, MA). Results were confirmed by at least three independent experiments. The antibodies used are listed in Supplementary Table 3.

### Co-immunoprecipitation (IP) assay

T-leukemic cells were lysed in IP buffer (Beyotime, China) and then centrifuged at 15000 × g for 30 min. Sample proteins were incubated with anti-Flag antibody (CST, Cat: 8146) at 4 °C for 8 h. Samples were washed with IP buffer 3 times and were incubated with in Protein-A Agarose for 4 h. Immunoprecipitants were washed with IP buffer 3

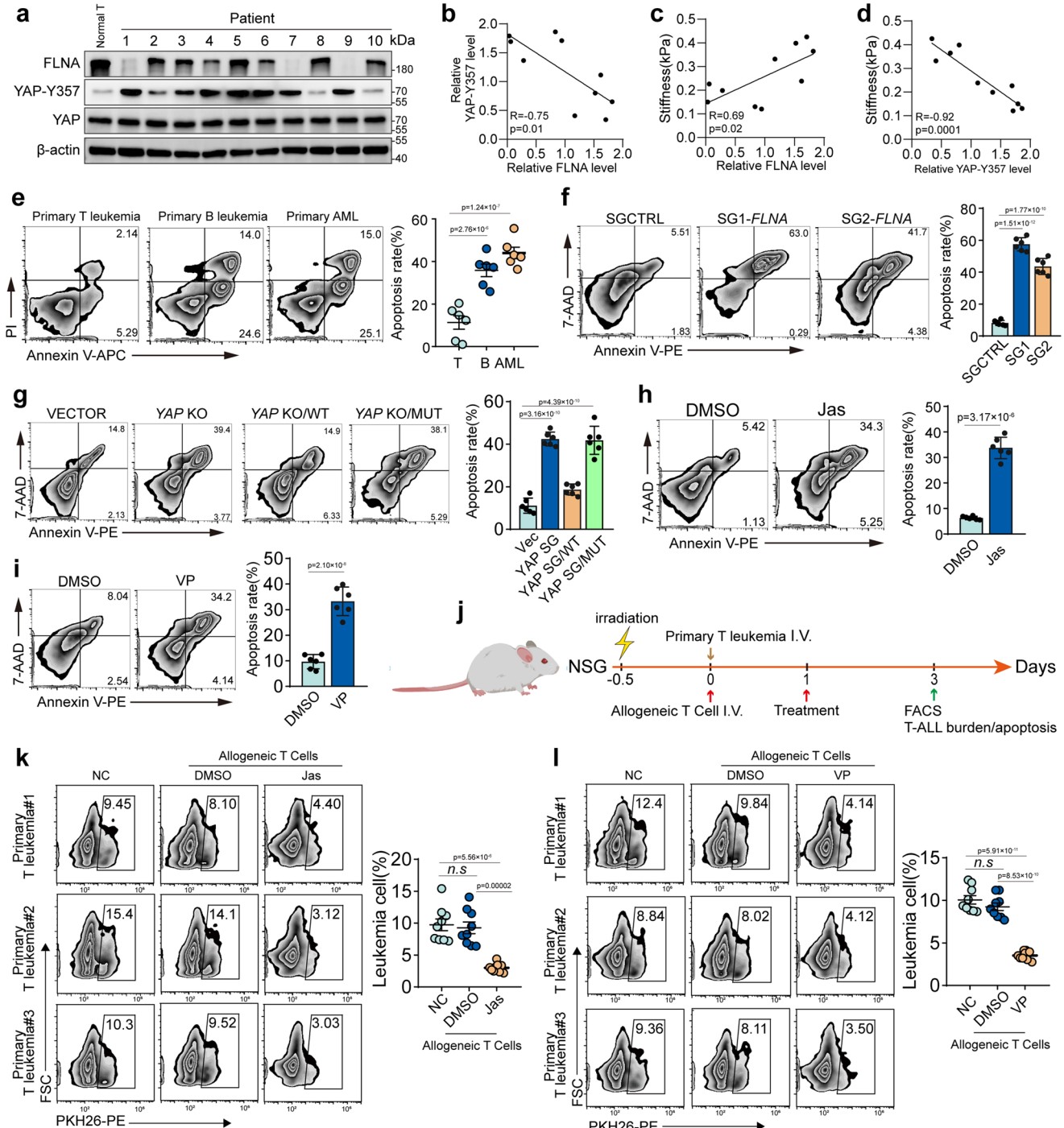

**Fig. 9 | Softness promotes T-leukemic cell immune evasion in patients. a** The expression of FLNA, YAP and Y357-YAP in human normal T cells or primary T leukemia cells were analyzed by western blot. **b** The correlation between the expression of Y357-YAP and FLNA was showed. **c** The correlation between the expression of FLNA and cell stiffness was showed. **d** The correlation between the expression of Y357-YAP and cell stiffness was showed. **e** The cell viability of the primary T leukemia cells, primary B-leukemic and primary acute myeloid leukemia cells was detected by flow cytometry after coculturing with allogeneic donor-derived T cells for 12 h. **f** The cell viability of SGCTRL or *FLNA*-SGs T-leukemic cells was detected by flow cytometry after coculturing with allogeneic donor-derived T cells for 12 h. **g** The cell viability of SGCTRL, *YAP* KO, *YAP* KO/*YAP* WT and *YAP* KO/ *YAP* MUT of T-leukemic cells was detected by flow cytometry after coculturing with allogeneic donor-derived T cells for 12 h. **h** The cell viability of T-leukemic cells treated with 200 nM Jas or DMSO was detected by flow cytometry after cocultur-

ing with allogeneic donor-derived T cells for 12 h. **i** The cell viability of T-leukemic cells treated 0.1 μM VP or DMSO was detected by flow cytometry after coculturing with allogeneic donor-derived T cells for 12 h. **j** Schematic of the experiment to test whether Jas or VP inhibits leukemia development in NSG mice. NSG mice were irradiated (2 Gy), then adoptively transferred with $1 \times 10^6$ primary leukemic T cells labeled with PKH26 and $1 \times 10^6$ allogeneic T cells. At 1 day post engraftment, NSG mice were treated with Jas (50 μg/kg) or VP (20 mg/kg). At 3 days post engraftment, PKH26$^+$ cells were analyzed by flow cytometry (Figure was created with BioRender.com). **k, l** Representative flow plots and quantification of percent of PKH26$^+$ primary T leukemia cells in bone marrow from mice treated with Jas (**k**) or VP (**l**). $n = 10$ biologically independent samples (**a**); $n = 6$ independent samples (**e**–**i**); $n = 9$ independent samples (**k** and **l**). The data are represented as mean ± SD. $p$ value by One-way ANOVA Bonferroni's test (**e**–**g** and **k**–**l**); $p$ value by two-tailed Student's $t$-test (**h** and **i**). Source data are provided as a Source Data file.

times and boiled in SDS sample buffer for 5 min. Then, the Immunoprecipitants were run on an SDS-PAGE gel.

## Isolation of CD8+ T cells

For Human, CD8+ T cells from healthy donors were isolated by RosetteSep (Stem Cell Technologies, Canada), activated with anti-CD3/CD28 beads and maintained in RPMI 1640 medium with 10% FBS and 50 U of rIL-2. For mouse, Ovalbumin (OVA) specific CD8+ T cells were isolated from the spleen of C57BL/6 mice, OT-I transgenic mice by negative selection using magnetic cell separation (MACS, mouse CD8+ T cell Isolation Kit, Miltenyi Biotec). Above 95% purity was confirmed by flow cytometry using anti-CD8 antibody (eBioscience, CA, USA). The isolated CD8+ T cells were cultured in RPMI 1640 medium containing 10% FBS and 100 U/ml IL-2 (eBioscience, CA) with activation by the anti-CD3/CD28 beads, OVA peptide (OVA$_{257-264}$, 2 μg/ml).

## Pore measurement by atomic force microscopy

The T cells and T-leukemic cells were cultured on the confocal dishes coated with RetroNectin (Takara Bio, CH-296, Otsu, Japan) (10 mg/ml) for 4 h according to the manufacturer instructions. Cells were pretreated with or without Jas or Lat-A for 12 h and then treated with recombinant perforin (rPRF) for 10 min. Cells were then washed 3 times with PBS and were fixed in 4% paraformaldehyde for 15 min. The formed pores were measured using a Dimension ICON AFM (Bruker, Santa Barbara, USA) set to PeakForce Tapping mode as described before[24]. The AFM probe is composed of a cantilever with a nominal spring constant of 0.1 Nm-1 and sharpened silicon tip (SCANASYST-AIR) with a nominal radius of 2 nm at its end. The cantilever is oscillated close to resonance while scanning across a sample. The tip only touches the sample at the very end of its downward movement thus considerably minimizing friction. In close proximity to the sample surface, the interactions between tip and sample change both the cantilever amplitude and resonance frequency allowing them to be used as feedback parameters for contouring fragile biological samples. The oscillating probe tips records series of force curves for each pixel points on the sample, where the maximum peak forces are maintained by tuning the vertical piezoelectric driver (z-piezo). The z-piezo movements of the probe, as a function of the x, y horizontal coordinates, are then be plotted as the sample 3D topography. The AFM topographs were recorded at room temperature and the maximum force applied to image the samples was 1 nN. The peak force frequency and peak force amplitude was set to 1 kHz and 300 nm, respectively. AFM images were analyzed and processed with Nanoscope Software (Bruker, Karlsruhe, Germany). Pore diameters were measured from both the major axis and short axis around the pore. We flattened each tapping mode image to measure the diameter and depth of every pore to eliminate the slope of the sample, but the depth of the pore in the sample itself does not change. A normalized scale of the images ranging from −100 nm to 100 nm was conducted to generate reliable data. Only when the shape of the pore was round or oval, and at the same time the depth of the pore is greater than 20 nm, it was considered as a real hole and not a normal fluctuation of the membrane. Pore diameters in each image was counted as average diameters.

Cell stiffness was measured by using a BioScope Resolve (Bruker, Santa Barbara, USA) AFM. The cantilevers for AFM were with a nominal spring constant of 0.5 Nm$^{-1}$. The AFM imaging were recorded at room temperature and the scan rate was 1.00 Hz, with tip velocity of 100 μm/s. The Peak Force Setpoint for Feedback was set to 0.6 V. The peak force frequency and peak force amplitude was set to 1 kHz and 300 nm, respectively. Hundreds of viable adherent tumor cells were imaged and the cellular stiffness was measured.

## Flow cytometry

For cell surface marker analysis, immune cells or leukemia cells were resuspended in PBS containing 1% FBS and stained with indicated fluorescent-conjugated antibodies for 30 min at 4 °C. Intracellular Fixation & Permeabilization Buffer Set (Invitrogen) was used according to the manufacturer's instructions. For PI staining, cells in HBSS solution with 2% BSA and 2 mM CaCl$_2$ were treated with perforin for the indicated time, and then performed the flow cytometry analysis. Data was analyzed with Flowjo software. The antibodies used are listed in Supplementary Table 3.

## Cell fractionation

The cytoplasmic and nuclear proteins were isolated by using the Cell Fractionation Kit (BioVision, SF, USA) according to the manufacture's instruction. Then, equal cell equivalents were analyzed by western blot.

## Immunofluorescence

T-leukemic cells were fixed in 4% paraformaldehyde and permeabilized with 0.2% Triton X-100. Fixed cells were blocked in 5% BSA and incubated with anti-YAP (Abcam, Cat.ab52771; 1:200), anti-FLNA (Abcam, Cat.76289; 1:200) at 4 °C overnight. Then, cells were washed and incubated with secondary antibodies for 1 h at room temperature. Finally, the slides were counterstained with DAPI or Phalloidin-iFluor (Abcam, Cat. 176753; 1:1000) and mounted for confocal analysis. The intensity of immunofluorescence was analyzed by Image J 9.0 software.

## In vitro kinase assay

In vitro kinase assays were performed as previously reported[73]. In brief, purified active Flag-ZAP70 (500 ng) was incubated with bacterially purified His−YAP1 (100 ng) in 25 μl kinase buffer (50 mM Tris-HCl at pH 7.5, 100 mM KCl, 50 mM MgCl$_2$, 1 mM Na$_3$VO$_4$, 1 mM DTT, 5% glycerol, 0.5 mM ATP) at 25 °C for 1 h. The reaction was terminated by adding SDS−PAGE loading buffer and heated at 100 °C for 5 min. The reaction mixture was then subjected to an SDS−PAGE analysis.

## Generation of CRISPR-Cas9 disruption or activation cell lines

For construction of the stable knockout of *YAP*-leukemic T cells, the YAP SGRNA primer sequences are listed in Supplementary Table 2. These SGRNAs were cloned into the pL-CRISPR.EFS.BFP vector plasmid. For construction of the stable activation of *FLNA*-leukemic T cells, the FLNA primer sequences are listed in Supplementary Table 2. These SGRNAs were cloned into the pKVL2-U6gRNA_SAM(BbsI)-PGK-puro-BFP-W vector plasmid (addgene# 112925). Lentivirus was produced by triple transfection of 293 T cells with a transfer vector, and the packaging plasmids psPAX2 and pMD.2 G at a 10:7:3 ratio. Transfection was performed using JetPEI reagent as recommended by the manufacturer. The viral supernatant was collected 48 h following transfection, filtered through a 0.45 μm filter, and added to target cells.

## CRISPR screening

Jurkat cells were transduced with a lentiviral vector encoding dCas9-V64-p65-HSF1 and selected with blasticidin. $1 \times 10^7$ cells Jurkat cells were infected with the pooled lentiviral Membrane protein gRNA library (addgene#113345) at a multiplicity of infection of 0.3 and selected with puromycin for 72 h, commencing 48 h after transduction. For the sorts, Jurkat cells were pre-treated with 100 ng/ml rPRF for 30 min, PI negative or positive cells were enriched by two rounds of FACS sorting following transduction with the sgRNA library. Genomic DNA was extracted (DNA Extraction Kits, Qiagen, Germany) from the sorted cells. sgRNA sequences were amplified by two rounds of PCR, with the second-round primers containing adaptors for Illumina sequencing. The resulting libraries were sequenced with single-end 50 bp reads on a HiSeq2500. The sequence reads were then aligned to gRNA sequences using the count function in MAGeCK. For testing of gene level enrichment, MAGeCK employs a modified Robust-Rank. The P value cut-off for

significance was adjusted to account for multiple-testing using the Bonferroni correction.

## Luciferase assays

293 T or NIH3T3 cells were transfected with 100 ng Renilla luciferase plasmid pRL-SV40, 1 μg firefly luciferase plasmid pGL4.10-FLNA or pGL4.10-FLNA mutant, and 1 μg of pCMVh-YAP or TEAD1 plasmid for 12 h. Cell lysates were analyzed using the Dual-Luciferase Reporter Assay (Promega, USA) on a GloMax Multi Plus (Promega, USA). Firefly luciferase activity was normalized to Renilla luciferase.

## ChIP PCR

ChIP was performed by using an iDeal ChIP-seq kit for Transcription factor (Diagenode, Belgium) according to the manufacturer's protocol. In brief, $1 \times 10^7$ cells were cross-lined with 1% formaldehyde for 8 min at room temperature. After stopping the fixation with 0.125 M glycine for 5 min at room temperature, cells were washed, lysed and sheared by sonication using Bioruptor (Diagenode, Belgium) for 20–30 cycles (30 s "ON", 30 s "OFF") at high power setting. The sheared chromatin fragments were immunoprecipitated with protein A-coated magnetic beads and anti-YAP (Cell Signaling, Cat.: 14074; 1:50); anti-TEAD1 (GeneTex, Cat.: GTX32918; 1:50); anti-NFAT1 (GeneTex, Cat.: GTX22722; 1:50) or anti-IgG (Cell Signaling, Cat.: 3900 S; 1:50) antibody. After elution and decross-linking, the enriched DNA fragments were purified by IPure beads v2. According to manufacturer's protocol. Specific ChIP-qPCR primer sequences are listed in Supplementary Table 2.

## Animal experiments and treatment protocol

For ICN-OVA/OT1 animal model, NSG mice were injected with $2 \times 10^5$ OVA-expressing ICN1 leukemic cells, at 7 days post engraftment, mice were randomized into different groups (sample size was $n = 6$) based on similar leukemia burden in peripheral blood and body weight. And then, NSG mice were treated or untreated with Jas(50 μg/kg/day), followed by transferring with or without leukemia specific CD8$^+$ T cells for the indicated time. The mice in the control groups received an equal volume of saline. For animal models of CAR-T therapy, NSG mice were transplanted with $3 \times 10^6$ YAP knockout and vector control Molt4 cells. At 7 days post engraftment, NSG mice were adoptively transferred with $1 \times 10^6$ human CD1a-CAR-T cells, followed by treating with cytoskeletal agents every 3 d, for a total of 3 times. At 30 days post engraftment, leukemia burden was measured by flow cytometry. For patient-derived xenograft models, 8-week-old NSG-B2M KO mice (GemPharmatech Co., Ltd, China.) were bred and housed under pathogen-free conditions in the animal facility of Institute of Basic Medical Sciences, Chinese Academy of Medical Sciences. Mice were irradiated (2 Gy) and transplanted with $1 \times 10^6$ primary leukemic T cells and $1 \times 10^6$ allogeneic T cell derived from health donor. And then, NSG mice were treated or untreated with Jas(50 μg/kg/day) for the indicated time.

To measure luminescence, mice were given 150 mg/kg of D-luciferin intraperitoneally, and tumor burden was monitored at the indicated time points. Living Image software (PerkinElmer) was used to visualize and calculate total luminescence. The leukemia growth was measured and long term survival was recorded.

For systemic toxicity analysis, the serum from NSG mice was separated for evaluation of creatinine (Cr), glutamic oxaloacetic transaminase (GOT), and glutamic pyruvic transaminase (GPT) in accordance with the protocols for the respective analysis kits (Nanjing Jiancheng Bioengineering Institute, China).

## Recombinant perforin

PRF and 6×His tag at the C-terminus was subcloned into pFastBac1 (presented by Dr. Feng Shao, National Institute of Biological Sciences, Beijing) to construct the recombinant plasmid of pFastBac1-PRF1.

Then, DH10Bac *E.coli* cells (provided by Dr. Feng Shao, National Institute of Biological Sciences, Beijing) were transformed with this plasmid to obtain a recombinant bacmid. Sf9 insect cells were transfected with this recombinant bacmid using Cellfectin II (Gibco, USA) according to the manufacturer's instructions. rPRF with C-terminal 6×His tag was purified from the supernatants of baculovirus infected Sf9 cells. The purity of the purified rPRF was assessed by electrophoresis on a 10% polyacrylamide gel and a 69 kDa band was visualized by silver staining and an immunoblot.

## Quantification and statistical analysis

All experiments were performed at least three times. Results are expressed as mean ± SD as indicated, and analyzed by Student's *t*-test followed by two-tailed Paired *t*-test or Mann–Whitney test, or one-way ANOVA followed by Bonferroni as indicated. The $p < 0.05$ was considered statistically significant. The analysis was conducted using GraphPad 8.0 software. The survival rates were evaluated by the Log-rank test.

## Reporting summary

Further information on research design is available in the Nature Portfolio Reporting Summary linked to this article.

## Data availability

The CRISPR screening data generated in this study have been deposited in the Genome Sequence Archive in National Genomics Data Center, Beijing Institute of Genomics (China National Center for Bioinformation), Chinese Academy of Sciences, under accession number HRA002823. Publicly available datasets in NCBI were used for Fig. 4 found at accession numbers: GSE26713. All the other data supporting the findings of this study are available within the article and its Supplementary Information files. Source data are provided with this paper.

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

## Acknowledgements

This work was supported by National Key Research and Development Program of China (2022YFA1206000, 2021YFC2500300), National Natural Science Foundation of China (82388201, 82300258, 82203549), Chinese Academy of Medical Sciences (CAMS) Innovation Fund for Medical Sciences (CIFMS) (2021-I2M-1-021), Haihe Laboratory of Cell Ecosystem Innovation Fund (22HHXBSS00009). The authors acknowledge the use of Biorender for creating schematic diagrams in Figs. 2a, 6a, 7a, and 9j, under the agreement number MI2674CMA4.

## Author contributions

B.H. conceived the project. Y.Z., D.W., L.Z., N.Z., Z.W., and J.C. performed the experiments. Y.Z., D.W., H.F., R.F., Q.H., and X.Z. developed methodology. Y.Z., D.W., R.P., F.D., H.C., H.Z., K.T., J.M., J.L., T.C., H.F., Q.H., and X.Z. performed data analysis. B.H. wrote the manuscript.

## Competing interests

The authors declare no competing interests.
