## [Peer Review File · Nature Communications]

Cell softness renders cytotoxic T lymphocytes and T leukemic cells resistant to perforin-mediated killingREVIEWER COMMENTS

Reviewer #1 (Anti-tumor immunity, CD8 biology) (Remarks to the Author):

This is a spectacular tour de force that may finally answer a question that has confounded immunologists for decades: what prevents a CTL from becoming lysed itself. The extension of the mechanistic explanation to the relative resistance of T-cell malignancies is an added bonus, as is the potential to pharmacologically enhance CTL anti-tumor function in vivo. All in all, this was a pleasure to review.

I have two minor comments that should be addressed.

1. Please clarify the scales shown in the AFM figures measuring pore size (eg, Fig 1h). What does "-135 nm" mean?
2. The authors state membrane softness at the immunological synapse protects CTLs from lysis. Perhaps I missed it, but this is inferred and not shown directly. That is, membrane softness at the immunological synapse was not measured directly; it was total membrane softness, correct? If so, then the wording/conclusions need to be modified accordingly.

Reviewer #2 (T cell activation, mechnobiology, imaging) (Remarks to the Author):

"Cell softness renders CTLs and T-leukemic cells resistant to perforin-mediated killing," by Zhou et al., examines the role of deformability in protecting CTLs and their cancer cell targets from perforin-mediated lysis. Using in vitro assays, the authors find that effector CTLs and leukemic T cells downregulate the F-actin crosslinking protein FLNA, thereby increasing their deformability and protecting them from perforin pore formation. This process is demonstrated to require YAP signaling, and the authors provide evidence that inhibiting YAP function or enforcing FLNA expression sensitizes leukemic cells to CTL-mediated killing both in vitro and in vivo. In doing so, they make a case for using YAP inhibitors at moderate doses as a strategy to enhance the efficacy of adoptive T cell immunotherapy. This study has an impressive scope and is quite interesting from a conceptual standpoint. In addition, the in vivo work combining YAP inhibition with CAR-T cells lays a foundation for future translational efforts. However, certain aspects of the approach, data analysis, and interpretation raise substantial concerns that must, in the opinion of this reviewer, be addressed before moving forward.

1) The AFM-based perforin pore formation assay is applied extensively throughout the manuscript, but the way it is described and presented is both confusing and concerning. All AFM panels feature representative images of each experimental group, but the scaling of these images is often dramatically different. In almost every case, images depicting samples with few or no pores are scaled to cover substantially more vertical distance than images depicting membranes with pores. For example, in supplementary figure S5d, the scaling for the three images is as follows: sg-Ctrl (no pores): 181.8nm, sg1-FLNA (pores): 70.8nm, sg2-FLNA (pores): 70.3nm. Similarly, for supplementary figure S5f, sg-Ctrl (no pores): 361.5nm, sg1-YAP (pores): 112nm, sg2-YAP: 96.4nm. This is the case for many, if not all, of the AFM panels. This pattern raises the obvious concern that the scaling of each image was tailored either to highlight the existence of pores (by shrinking the scaling, thereby emphasizing variation) or to make them less apparent (by expanding the scaling, thereby smoothing things out). Needless to say, comparing images with such different scaling is essentially meaningless. Importantly, if more appropriate scaling were used, would the authors' conclusions change?

The image quantitation is also opaque. In the methods section, it is stated that the authors “flattened each tapping mode image to measure the diameter and depth of every pore.” How can pore depth be measured from a flattened image?

Furthermore, it is unclear what the pores visualized in their AFM images actually represent. Perforin pores have been documented to be 5-30 nm in diameter. The pores shown in the representative images seem much larger than this, as much as an order of magnitude larger. Are they really perforin pores, or just regions of membrane damage/remodeling? More must be done to characterize these structures.

Also unclear is how the authors selected which images to collect and quantitate. What efforts were made to ensure a representative sampling of the cells? Were entire cells probed or just the 2 micron by 2 micron squares exemplified by the representative images? If only 2 micron by 2 micron squares were collected, then how were these specific regions chosen?

The authors need to clarify the presentation of their AFM data in the methods, legends, and the figure panels themselves. As it stands, the AFM data in this manuscript falls well short of being convincing.

2) Using Jasplakinolide for in vivo studies (e.g. Fig. 5) will disturb F-actin architecture in all cells, likely generating all manner of nonspecific effects. I am surprised that T cells can even traffic to tumors under these conditions. The same concern applies to the use of Jasplakinolide in the in vitro killing assay shown in Fig 7g. The T cells will certainly be affected. The authors must do more to establish the mechanism for synergy between therapeutic T cells, Jasplakinolide, and perforin, or remove the Jasplakinolide experiments from the manuscript altogether.

3) It is surprising that the authors can recover PI+ cells from a mouse (Fig. 5b). If these cells are perforated and undergoing programmed cell death, one would expect that they would be rapidly cleared. Perhaps Jasplakinolide inhibits efferocytosis by patrolling phagocytes?

4) For the CRISPRa screen (Fig. 2), what fraction of Jurkat cells become PI positive upon perforin treatment in the absence of gRNA library transfection? I'm trying to get a sense of how likely it would be to find a gene whose upregulation actually sensitizes cells to perforin using this assay structure.

5) The observation that effector T cells have less F-actin (Fig. 2g) runs counter to Thauland et al., *Sci Signaling* eaah3737. What do the authors think is going on here? Is it a cell type specific difference? Might the results depend on culture conditions?

6) The heatmap in Fig. 4a is confusing. The legend refers to 200 “highly expressed” genes in T-ALL, yet all genes appear to be much more weakly expressed in T-ALL than in bone marrow cells. The comparison with normal bone marrow is also perplexing. Wouldn't it be better to compare T-ALL cells to a T cell subset?

7) The data in Fig 6a-d do not necessarily support a model in which YAP deficiency enhances T cell mediated killing of tumor cells. The effects look additive, rather than synergistic, indicating that YAP and the CAR T cells could be inhibiting tumor growth by independent mechanisms.

8) Fig. 6j appears to indicate that treatment of T cells with low-dose verteporfin induces IFN γ and TNF on its own. Is this the case? The figure legend states that the T cells were “activated”. Does this mean that they were stimulated through the TCR, as well?

9) In Fig. S8k and Fig. 7j-k, what is the significance of the “PKH26” channel? Do the authors mean PKH26, a cell staining dye that fluoresces in the PE channel? If so, does this mean that they used PKH26 to label leukemic cells before adoptive transfer in vivo? Is it reasonable to expect that this dye will persist the entire duration of their experiment (3 days)?

10) Since the authors have access to 10 primary leukemias, each with a different level of FLNA expression (Fig. 7a), it would be interesting to compare their sensitivity to T cell killing, to see whether that sensitivity scaled with FLNA content.

11) A number of confusing statements are made in the Discussion. A four step model for CTL resistance to perforin is proposed, but the justification for steps 2-4 is not apparent to me. All three steps pertain to the relationship between the cytoskeleton and lipid order in the plasma membrane. This manuscript is a strange venue for showcasing this model, given that none of the presented data actually addresses its key aspects.

The idea that FLNA is mechanically coupled to perforin through the F-actin cytoskeleton is also very poorly supported. There is no law against the authors proposing this, of course, but other explanations related to cortical stiffening, changes in membrane tension, or changes in membrane curvature seem more reasonable at this stage.

Finally the authors imply that Zap70 directly phosphorylates YAP, but their results provide no evidence that this phosphorylation is direct, only that Zap70 is required for YAP phosphorylation. This effect could very well be secondary.

Reviewer #3 (CTL responses, anti-tumor immunity) (Remarks to the Author):

Drs Zhou, Huang and colleagues report that “cell softness”, in addition to its previously identified role in protecting CTLs from accidental self-killing, is also used by T-leukemic cells to resist the perforin-mediated killing. The manuscript includes an impressive amount of data from human and mouse cell cultures and mouse in vivo experiments in support of the scenario where the downregulation of filamin A (FLNA) enhances cell softness, mediated by YAP activation, driven in turn by Y357 phosphorylation by ZAP70. The authors report that CTLs are more resistant to YAP inhibitors than malignant T cells, due to the high expression of MDR1 in CTLs. As a result, low level inhibition of YAP enhances stiffness in T cells but not CTLs allowing CTLs to safely kill the malignant cells. The overall quality of the experimental work is very high.

There are two significant areas of major concern:

1. The key novel message of the paper that CTLs are less susceptible to YAP inhibition than T-ALL cells and that this difference results in their different reliance on cell softness lacks definitive data support. First, the current experiments seem to compare the impact of perforin on a human and a mouse long-term T-ALL cell lines (ICN1 and 18 Molt4) with freshly-isolated and short-term-cultured primary T cells. The notion of a differential regulation OF YAP and FLNA system is supported by the genomic analysis of previously banked 117 T-

ALL samples and 7 normal bone samples in Figure 4, but in the absence of the direct comparison of fresh samples in functional studies, it is difficult to validate the central conclusion.

2. Due to the multitude of effects of the YAP-TEAD system on cancer cells, I am not sure if the currently presented data allows to conclude that the enhanced (although to a modest level) antitumor effects of CD1a CAR-T cells in the presence of YAP inhibition are indeed mediated by changes in the cell softness.

Minor comments:

3. The current figures provide a very significant amount of non-essential confirmatory data (such as ability of perforin to permeabilize cancer cells), which dilute the novel aspects of the paper and reduce its appeal. I also suggest to separate the human and mouse aspects of the paper, which are currently often presented in the same figures and display panels, reducing the clarity of the presentation.

4. The abstract and introduction could benefit from additional discussion of the specific novel questions, compared to the previously reported conclusions. The paper could also benefit from the additional discussion of how the results can help to modify the current clinical testing of YAP-TEAD inhibitors.

INSTITUTE OF BASIC MEDICAL SCIENCES, CHINESE ACADEMY OF MEDICAL SCIENCES

National Key Laboratory of Medical Molecular Biology & Department of Immunology
Dong Dan San Tiao, No.5
100005, Beijing, China

RESPONSES TO REVIEWERS

We would like to express our sincere thanks to three reviewers for their critical and constructive comments. We respond point-by-point to each of their comments and criticisms. We feel that their comments have helped us on substantially improving and strengthening the manuscript and clarifying some issues.

REVIEWER COMMENTS

Reviewer #1 (Anti-tumor immunity, CD8 biology) (Remarks to the Author):

This is a spectacular tour de force that may finally answer a question that has confounded immunologists for decades: what prevents a CTL from becoming lysed itself. The extension of the mechanistic explanation to the relative resistance of T-cell malignancies is an added bonus, as is the potential to pharmacologically enhance CTL anti-tumor function in vivo. All in all, this was a pleasure to review.

I have two minor comments that should be addressed.

1. Please clarify the scales shown in the AFM figures measuring pore size (eg, Fig 1h).
What does "-135 nm" mean?

Response:

We thank the reviewer's suggestion. In this study, we used the PeakForce Tapping mode AFM to measure perforin pore formation, as described previously (Cell Mol Immunol. 2019 Jun;16(6):611-620.). The pore information was collected in a $2 \times 2 \mu\text{m}^2$ area. We performed a metrological analysis of the diameter (long and short lengths from the horizontal surface) and depth (the maximum vertical distance between the highest and lowest data points) of pore-like structures in samples using AFM NanoScope Analysis 1.9 software. The probe assigns a value to the position of each point on the cell surface. These values are then normalized, with the average height selected as the initial plane (0 nm). When the probe shows a negative value, the lower it is the darker it is within the image. "-135nm" means that the probe is at the position 135nm below the initial plane.

2. The authors state membrane softness at the immunological synapse protects CTLs from lysis. Perhaps I missed it, but this is inferred and not shown directly. That is, membrane softness at the immunological synapse was not measured directly; it was total membrane softness, correct? If so, then the wording/conclusions need to be modified accordingly.

Response:

We thank the reviewer's suggestion. Stiffness does not refer to that of a cellular membrane. Cell plasma membrane stiffness is known to be extremely low, on the order of ~ 0.1 - 1 Pa (Waugh R, Evans EA. Thermoelasticity of Red Blood-Cell Membrane. *Biophys J.* 1979;26:115-31). The stiffness measured by AFM is a property of the membrane cortex that includes the plasma membrane plus the underlying tensed cytoskeleton (F-actin and myosin II, microtubules, etc). Therefore, the rigidity of the plasma membrane is negligible compared to that of the cytoskeleton (*Integr. Biol.* 7, 356–363 2015.). In this study, when we mentioned the softness at the synapse, it was the local plasma membrane plus the underlying tensed cytoskeleton.

Reviewer #2 (T cell activation, mechnobiology, imaging) (Remarks to the Author):

“Cell softness renders CTLs and T-leukemic cells resistant to perforin-mediated killing,” by Zhou et al., examines the role of deformability in protecting CTLs and their cancer cell targets from perforin-mediated lysis. Using in vitro assays, the authors find that effector CTLs and leukemic T cells downregulate the F-actin crosslinking protein FLNA, thereby increasing their deformability and protecting them from perforin pore formation. This process is demonstrated to require YAP signaling, and the authors provide evidence that inhibiting YAP function or enforcing FLNA expression sensitizes leukemic cells to CTL-mediated killing both in vitro and in vivo. In doing so, they make a case for using YAP inhibitors at moderate doses as a strategy to enhance the efficacy of adoptive T cell immunotherapy. This study has an impressive scope and is quite interesting from a conceptual standpoint. In addition, the in vivo work combining YAP inhibition with CAR-T cells lays a foundation for future translational efforts. However, certain aspects of the approach, data analysis, and interpretation raise substantial concerns that must, in the opinion of this reviewer, be addressed before moving forward.

- 1) The AFM-based perforin pore formation assay is applied extensively throughout the manuscript, but the way it is described and presented is both confusing and concerning. All AFM panels feature representative images of each experimental group, but the scaling of these images is often dramatically different. In almost every case, images depicting samples with few or no pores are scaled to cover substantially more vertical distance than images depicting membranes with pores. For example, in supplementary figure S5d, the scaling for the three images is as follows: sg-Ctrl (no pores): 181.8nm, sg1-FLNA (pores): 70.8nm, sg2-FLNA (pores): 70.3nm. Similarly, for supplementary figure S5f, sg-Ctrl (no pores): 361.5nm, sg1-YAP (pores): 112nm, sg2-YAP: 96.4nm. This is the case for many, if not all, of the AFM panels. This pattern raises the obvious concern that the scaling of each image was tailored either to highlight the existence of pores (by shrinking the scaling, thereby emphasizing variation) or to make them less apparent (by expanding the scaling, thereby smoothing things out). Needless to say, comparing images with such different scaling is essentially meaningless. Importantly, if more appropriate scaling were used, would the authors' conclusions change?

Response:

We thank the reviewer's comment. The basic AFM configuration contains four parts: a probe as one complete cantilever with a sharp tip mounted at its end, a piezo-actuator that drives the probe, a laser source, and a position sensitive detector (PSD). As the probe tip scans a sample in close proximity, the quantitative mechanical interactions between the probe tip and sample surface are measured by PSD signals, and the sample 3D topography is plotted by recoding the z-piezo movements of the tip according to the x,y horizontal coordinates. Based on this principle, we have developed the use of AFM technology to measure perforin pore formation in true cells in the lab (Cell Mol Immunol. 2019; 16:611-20; Sci Immunol. 2020;5: eaax7969; Cancer Res. 2021; 81:476-88; Front Med. 2021;15:43-52). This method uses the Peak Force Tapping mode AFM in that probe tip does not contact the sample surface continuously but intermittently and with a fixed oscillation frequency.

Pores formed in the plasma membrane can be characterized by steep trenches in the surface morphology. PeakForce Tapping mode AFM is nicely adapted to this situation with very low probe forces and high imaging resolution since it is insensitive to the geometric effects and has no difficulty probing the bottom of trenches. We used the software of NanoScope Analysis 1.8 to analyze the images taken by AFM. By using section analysis supplied by the software, which is usually used for analysis of roughness of surface, we can see the recorded curve reflecting the moving track of the tip. The images showed U-type curves with a flat bottom, indicating that a pore structure is detected by the tip, which can be precisely illustrated as the size and depth of formed pores. In our experiments, the PeakForce Tapping mode AFM was equipped with a 90 μ m piezoelectric scanner (0.4 N/m for nominal spring constant) and a sharpened silicon tip (2 nm for nominal radius) in an acoustic isolation box. In addition, the cantilever oscillation frequency, oscillation amplitude, and the maximum probe force during AFM measurements were set to be 2 kHz, 50nm and 1nN, respectively. As long as we kept the same parameters under the consistent condition, the results should be stable and convincing. In addition, we normalized the uniformity of the scale bars using AFM analysis software. Then, we carried out a statistical analysis, and the results were consistent with our previous results.

In addition to AFM, the measurement of pore formation also relies on cellular state. Pore formation in the plasma membrane is highly dynamic. Cells have the ability to repair membrane pores rapidly (Cell Mol Immunol. 2019; 16:611-20). Different batches of T cells are under different states, which may result in differences in pore repairment outcomes and measured pore sizes. Despite this difference, consistent trends were observed among different batches, when the membrane pores were measured in T cells.

The image quantitation is also opaque. In the methods section, it is stated that the authors "flattened each tapping mode image to measure the diameter and depth of every pore."

How can pore depth be measured from a flattened image?

Furthermore, it is unclear what the pores visualized in their AFM images actually represent.

Perforin pores have been documented to be 5-30 nm in diameter. The pores shown in the representative images seem much larger than this, as much as an order of magnitude larger. Are they really perforin pores, or just regions of membrane damage/remodeling? More must be done to characterize these structures.

Response:

We thank the reviewer's comment. During imaging of the samples by atomic force microscopy, cells, which adhered the surface of culture dish, were captured with a spherical shape. When the probe contacted cell surface, there were inclines in the sampling areas, which affected the imaging of pores (please see Figure A below). By flattening the image while maintaining the shape and depth of the pores, we obtained a clearer and more realistic image (please see Figure B below). We have not added this information to the revised manuscript to make the writing more concise and smoother.

We understood the reviewer's concern on the pore size. Previously documented pore measurement was based on liposome membrane, which showed 5-30 nm in diameter (Sci Adv. 2022;8(6):eabk3147.). When perforin forms pores in liposomes, it only interacted with lipid molecules. However, when perforin forms pores in real plasma membrane, it interacts with both lipid molecules and protein molecules, thus resulting in different pore sizes. Liposomal pores cannot reflect the size of pores in true plasma membranes. Notably, perforin pores in the plasma membrane are very dynamic, which can undergo fusion and enlargement over time with perforin activity. In our previously experiments, we captured the dynamic nature of perforin pore formation in live cells, and the diameter of the pores could be as large as several hundred nanometers (Cell Mol Immunol. 2019; 16:611-20; Sci Immunol. 2020;5: eaax7969; Cancer Res. 2021; 81:476-88). It is an automatic processing to flatten image in the NanoScope Analysis 1.9 software, which transforms pores on a slope into a horizontal plane pore, making it easier to measure height differences (Nat Commun. 2022 Aug 26;13(1):5039).

Figure, (A) High-resolution AFM topography raw data was performed and the pores were analyzed by NanoScope Analysis 1.9. (B) The high-resolution AFM morphology raw data were flattened and the pores was analyzed by NanoScope Analysis 1.9.

Also unclear is how the authors selected which images to collect and quantitate. What efforts were made to ensure a representative sampling of the cells? Were entire cells probed or just the 2 micron by 2 micron squares exemplified by the representative images? If only 2 micron by 2 micron squares were collected, then how were these specific regions chosen?

Response:

We thank the reviewer's comment. In this study, we randomly selected three locations in the plasma membrane of a cell to image the pores. The pore information was collected in a $2 \times 2 \mu\text{m}^2$ area in order to accurately obtain pores information and a clear field of view. This process was repeated in 10 cells to avoid variation. The representative images shown in the figures were chosen to reflect the observed trends and patterns. The raw data on measuring pores with AFM reported in this paper have been deposited in the Science Data Bank (<https://www.scidb.cn/en/s/Uzei22>). Moreover, the uncropped image on measuring pores with AFM reported in this paper have been provided in source data.

The authors need to clarify the presentation of their AFM data in the methods, legends, and the figure panels themselves. As it stands, the AFM data in this manuscript falls well short of being convincing.

Response:

We thank the reviewer's comment. In the revised manuscript, we have added the clear information to clarify the presentation of the AFM data in the methods, legends, and the figure panels. These modifications should make the AFM data convincing.

2) Using Jasplakinolide for in vivo studies (e.g. Fig. 5) will disturb F-actin architecture in all cells, likely generating all manner of nonspecific effects. I am surprised that T cells can even traffic to tumors under these conditions. The same concern applies to the use of Jasplakinolide in the in vitro killing assay shown in Fig 7g. The T cells will certainly be affected. The authors must do more to establish the mechanism for synergy between therapeutic T cells, Jasplakinolide, and perforin, or remove the Jasplakinolide experiments from the manuscript altogether.

Response:

We thank the reviewer's comment. Previous studies, including our own results, have found that effector T cells express higher levels of MDR1 compared to tumor cells, which enables T cells to survive and exert their cytotoxic effects in hostile tumor microenvironments (Please see figure below). We found that treatment with 50 $\mu\text{g}/\text{kg}$ Jasplakinolide in mice did not increase cellular stiffness in OT-1 T cells, which, however, stiffened T leukemia cells, suggesting that the administration of Jasplakinolide does not have a transmissible effect on the mechanical properties of OT-1 T cells (Please see figure below).

Figure (A) The stiffness of ICN cells or CTLs, isolated from Jas-treated mice, was determined by AFM. All error bars are mean \pm SD, p values were calculated by one-way ANOVA Bonferroni's test, ***p < 0.001.

We have not added this information to the revised manuscript to make the writing more concise and smoother.

3) It is surprising that the authors can recover PI+ cells from a mouse (Fig. 5b). If these cells are perforated and undergoing programmed cell death, one would expect that they would be rapidly cleared. Perhaps Jasplakinolide inhibits efferocytosis by patrolling phagocytes?

Response:

Tumor cell death is a common phenomenon in tumor mass, due to hypoxia, nutrient deprivation, radiation, chemotherapy and/or immuno-killing. Such dead tumor cells are usually observed by multiple methods including PI staining (Cold Spring Harb Protoc;2014(11):1202-6.). Tumor has been depicted as wounds that never heal, which may explain the continuous tumor cell death.

In this study, we used NSG mice (NOD.Cg-Prkdcscid Il2rgtm1Wjl/SzJ) for the experiments. After 48 hours of transferring OT-1 cells into NSG mice bearing ICN, we detected apoptosis of ICN immediately. At this time point, the leukemia cells, which were stiffened by Jasplakinolide, were efficiently killed by OVA-specific T cells, thus allowing us to observe PI-positive cells, as shown in the original Figure 5a-b. Moreover, in these mice, not only T cells, B cells and NK cells are deficient but macrophages and dendritic cells exhibit impaired phagocytosis (Int J Mol Sci. 2022; 23(9): 4680.J Immunol. 1995;154(1):180-91.). Therefore, as shown in Fig. 5b, we could observe the PI+ cells.

4) For the CRISPRa screen (Fig. 2), what fraction of Jurkat cells become PI positive upon perforin treatment in the absence of gRNA library transfection? I'm trying to get a sense of how likely it would be to find a gene whose upregulation actually sensitizes cells to perforin using this assay structure.

Response:

The CRISPRa system is designed to screen target genes by activating the target gene promoter and overexpressing target genes. When we used 50 ng/ml perforin to treat Jurkat cells, we found that more than 90% cells were resistant (negative PI staining). Intriguingly, the introduction of CRISPRa gRNA library into Jurkat cells markedly increased the cellular sensitivity to perforin (positive PI staining). By analyzing differentially expressed sgRNAs

between PI+ and PI- Jurkat cells, we found that filamin A (FLNA) was among the top 10 hits. Thus, we identified that FLNA was a candidate that may contribute to perforin pore formation.

5) The observation that effector T cells have less F-actin (Fig. 2g) runs counter to Thauland et al., Sci Signaling eaah3737. What do the authors think is going on here? Is it a cell type specific difference? Might the results depend on culture conditions?

Response:

The study by Thauland et al. showed that naïve T cells had a mechanically stiffer cortical cytoskeleton than that of effector cells (please see the Fig.1 and Fig. 2 of the article). This was consistent with our findings. In addition, studies by Govendir et al. (Dev Cell. 2022; 57:2237-2247.e8) showed that T cell F-actin was cleared at the synaptic region, which resulted in negative membrane curvatures. These findings were also consistent with our results.

6) The heatmap in Fig. 4a is confusing. The legend refers to 200 “highly expressed” genes in T-ALL, yet all genes appear to be much more weakly expressed in T-ALL than in bone marrow cells. The comparison with normal bone marrow is also perplexing. Wouldn't it be better to compare T-ALL cells to a T cell subset?

Response:

We thank the reviewer's indicating this writing error. We corrected it as “lowly expressed” in the revised manuscript. This result was derived from public database, in which gene expression data were available in T-ALL and bone marrow cells. This was why we did not use T cell subset as a comparison.

7) The data in Fig 6a-d do not necessarily support a model in which YAP deficiency enhances T cell mediated killing of tumor cells. The effects look additive, rather than synergistic, indicating that YAP and the CAR T cells could be inhibiting tumor growth by independent mechanisms.

Response:

In this study, we first demonstrated that YAP downregulates FLNA expression in T leukemic cells, leading to softening cells and impeding T cell-mediated killing. In Fig. 6a-d, we used YAP-deficient Molt4 T leukemic cells as target cells to validate that inhibition of target cell's YAP is useful to enhance T cell immunotherapy in vivo. We found that the combination of CAR-T cells and YAP KO generated much better treatment outcome, compared to single treatment groups (please see the original Fig. 6c), suggesting that the result in Fig. 6a-d is not additive. Such results raised a question of how to allow the inhibition of YAP in tumor cells but avoid the inhibition of YAP in T cells during T cell immunotherapy in vivo. We then addressed the question in the following experiments (Fig. 6e-i).

In the revised manuscript, we repeated the original CAR-T experiment using a lower effector-to-target ratio (CD1a CAR-T cell/ YAP1 KO Molt4-luc cell = 1:3). The result showed that the control sgCTRL cells were weakly destroyed; in contrast, YAP1 KO cells were

highly destroyed by CAR-T cells, concomitant with prolonged survival of the NSG mice (Please see figure below). However, this sensitivity of YAP1 KO cells was attenuated by treatment with latrunculin A, which inhibits F-actin and softens cells. According to the reviewer's comment, we added these new results in the revised manuscript, page 10 line 216-219, fig 6a-c and S7a-c.

Moreover, we adoptively co-transferred T-leukemic cells isolated from patients and CD8+ T cells isolated from healthy donors into irradiated NSG mice. We found that the T-leukemic cells were minimally affected by allogeneic T cells; however, YAP knockout conferred the sensitivity of T-leukemic cells to allogeneic T cells. Notably, such YAP1 knockout-promoted cytotoxicity was abrogated by latrunculin A or FLNA/YAP1 dual knockout. According to the reviewer's comment, we added these new results in the revised manuscript, page 12 line 265-268, fig S8g-h.

Figure, (A-B) NSG mice were transplanted with 3×10^6 YAP knockout and vector control Molt4-luc cells. Seven days after transplantation, NSG mice were adoptively transferred with 1×10^6 human CD1a-CAR-T cells. Thirty days later, leukemia burden (spleen and bone marrow) was analyzed by flow cytometry. (C) the same as (A-B), the survival was analyzed. (C) the same as (D-E), the survival was analyzed. (D-E) the same as (A-B), except that NSG mice were treated with PBS or Lat-A. (G-H) NSG mice were irradiated (2

Gy), then adoptively transferred with 1×10^6 PKH26-labeled primary leukemic T cells (transfected with vector, YAP1 KO and YAP1/FLNA KO) and 1×10^6 allogeneic T cells. One day later, NSG mice were treated with PBS or Lat-A (20 ug/kg). On day 3, PKH26⁺ cells were analyzed by flow cytometry. Representative flow plots and quantification of percent of PKH26⁺ primary T leukemia cells in bone marrow from mice. All error bars are mean \pm SD, n.s., not significant for $p > 0.05$, ** $P < 0.01$, *** $P < 0.001$, p values were calculated by one-way ANOVA Bonferroni's test (B, E, G, H) or Log-rank test (C, F), *** $p < 0.001$.

8) Fig. 6j appears to indicate that treatment of T cells with low-dose verteporfin induces IFN γ and TNF on its own. Is this the case? The figure legend states that the T cells were "activated". Does this mean that they were stimulated through the TCR, as well?

Response:

Previous studies have demonstrated that YAP inhibition enhances T cell activation (PLoS Biol. 2020 Jan 13;18(1): e3000591; refxxx). In this study, we also found that the YAP inhibitor verteporfin augmented T cell activation. YAP is poorly expressed in naïve T cells and the low-dose verteporfin did not induce IFN- γ and TNF in the T cells (Please see figure below). For Fig. 6j, T cells were stimulated with PMA rather than the use of TCR signaling.

Figure, (A) Naïve and activated murine CD8⁺ T cells were treated with 0.1 μ M VP for 24 h. qPCR analysis of the mRNA expression of IL-2, TNF α , Granzyme B, Perforin and IFN γ . All error bars are mean \pm SD, p values were calculated by one-way ANOVA Bonferroni's test, *** $p < 0.001$.

We did not add the showed this figure to the revised manuscript in order to make the writing more logical and fluent.

9) In Fig. S8k and Fig. 7j-k, what is the significance of the "PKH26" channel? Do the authors mean PKH26, a cell staining dye that fluoresces in the PE channel? If so, does this mean that they used PKH26 to label leukemic cells before adoptive transfer in vivo? Is it reasonable to expect that this dye will persist the entire duration of their experiment (3 days)?

Response:

PKH26 is a lipophilic fluorescent dye that can stably bind to the lipid molecules in the cell membrane, allowing to label live cells and their membrane structures. By emitting red fluorescence, PKH26 is detected in the PE channel. Cells labeled with PKH26 exhibit high stability and low toxicity, making it suitable for in vivo tracking. It has been reported that

PKH26 can persist in vivo for more than 49 days (Tissue Cell. 2012 Jun;44(3):156-63). In this study, we co-transferred PKH26-labeled leukemic cells and allogeneic T cells into YAP inhibitor or Jas-treated mice to track the survival of PKH26+ leukemic cells.

10) Since the authors have access to 10 primary leukemias, each with a different level of FLNA expression (Fig. 7a), it would be interesting to compare their sensitivity to T cell killing, to see whether that sensitivity scaled with FLNA content.

Response:

We thank the reviewer's constructive suggestion. It is undoubtedly helpful to validate FLNA expression levels related to T cell killing in the 10 primary leukemias. In the revised manuscript, we performed the cytotoxicity assay on the 10 primary leukemias using allogeneic T cells as effector cells, and found that FLNA and phosphorylated YAP1 (Y357) were positively correlated to the killing rate (please see figure below).

Figure, (A-B) Cell viability of the primary T leukemia cells, primary B-leukemic and acute myeloid leukemic cells was detected by flow cytometry after coculturing with allogeneic T cells for 12 hr. The correlation between FLNA (A) and phosphorylated YAP1 (Y357) (B) and apoptosis rate was showed (n=10). p values were calculated by Pearson's correlation test (A-B).

According to the reviewer's comment, we added these new results in the revised manuscript, page 12 line 258-260, fig S8b, c.

11) A number of confusing statements are made in the Discussion. A four step model for CTL resistance to perforin is proposed, but the justification for steps 2-4 is not apparent to me. All three steps pertain to the relationship between the cytoskeleton and lipid order in the plasma membrane. This manuscript is a strange venue for showcasing this model, given that none of the presented data actually addresses its key aspects.

The idea that FLNA is mechanically coupled to perforin through the F-actin cytoskeleton is also very poorly supported. There is no law against the authors proposing this, of course, but other explanations related to cortical stiffening, changes in membrane tension, or changes in membrane curvature seem more reasonable at this stage.

Response:

We thank the reviewer's comment. In general, cell stiffness includes cortical stiffness, cytoplasmic stiffness and nuclear stiffness. When we use AFM to measure cell stiffness, it

does not include the membrane tension and curvature, which have to be determined using different methods. It should clarify that the stiffness of plasma membrane is extremely low (Biophys J. 1979;26:115-31), and does not impact whole cell stiffness. We tried to use the four-step model to sketch the resistance of autologous T cells to perforin. However, to avoid confusion, mentioned by the reviewer, we remove this model from the manuscript.

Based on our findings and others, we propose that activated T cells mobilize lytic granules and allow the release of perforin and granzymes into the immune synaptic space via membrane fusion, resulting in perforin pore formation in stiff target cells but not in soft autologous T cells.

Finally, the authors imply that Zap70 directly phosphorylates YAP, but their results provide no evidence that this phosphorylation is direct, only that Zap70 is required for YAP phosphorylation. This effect could very well be secondary.

Response:

We thank the reviewer's constructive comment. We purified YAP from E. coli and active ZAP70 from Jurkat cells. By conducting the in vitro kinase assay, we found that YAP was directly phosphorylated by ZAP70, which, however, was abrogated by the addition of lambda protein phosphatase (Please see Figure below).

Figure, (A) Purified recombinant human YAP1 was incubated with active ZAP70 for in vitro kinase assay. Phosphorylated and total YAP1 and ZAP70 were determined by western blot.

According to the reviewer's suggestion, we added this result in the revised manuscript, page 8 line 176-178, fig 4m.

Reviewer #3 (CTL responses, anti-tumor immunity) (Remarks to the Author):

Drs Zhou, Huang and colleagues report that “cell softness”, in addition to its previously identified role in protecting CTLs from accidental self-killing, is also used by T-leukemic cells to resist the perforin-mediated killing. The manuscript includes an impressive amount of data from human and mouse cell cultures and mouse in vivo experiments in support of the scenario where the downregulation of filamin A (FLNA) enhances cell softness, mediated by YAP activation, driven in turn by Y357 phosphorylation by ZAP70. The authors report that CTLs are more resistant to YAP inhibitors than malignant T cells, due to the high expression of MDR1 in CTLs. As a result, low level inhibition of YAP enhances stiffness in T cells but not CTLs allowing CTLs to safely kill the malignant cells. The overall quality of the experimental work is very high.

There are two significant areas of major concern:

1. The key novel message of the paper that CTLs are less susceptible to YAP inhibition than T-ALL cells and that this difference results in their different reliance on cell softness lacks definitive data support. First, the current experiments seem to compare the impact of perforin on a human and a mouse long-term T-ALL cell lines (ICN1 and 18 Molt4) with freshly-isolated and short-term-cultured primary T cells. The notion of a differential regulation OF YAP and FLNA system is supported by the genomic analysis of previously banked 117 T-ALL samples and 7 normal bone samples in Figure 4, but in the absence of the direct comparison of fresh samples in functional studies, it is difficult to validate the central conclusion.

Response:

We thank the reviewer's constructive comment. In the original manuscript, we compared the levels of FLNA and phosphorylated YAP between primary CD8+ T cells and leukemic cells, both of which were isolated from the same patient (n=10) (please see original figure 7a). In the revised manuscript, we quantitated the western blot (phosphorylated YAP1 and FLNA) of the 10 samples. Moreover, we analyzed the expression of MDR1 in the 10 samples (please see figure A below). We found that activated primary T cells expressed higher levels of MDR1, compared to unstimulated counterparts or leukemic cells (please see figure B below). We then used the YAP inhibitor verteporfin to treat cells and analyzed verteporfin content in the cells using high performance liquid chromatography (HPLC). As expected, we found that verteporfin levels were much higher in leukemia cells than in activated T cells, concomitant with increased stiffness of leukemic cells rather than activated CD8+ T cells (please see figure C below). These results further suggest that effector T cells rather than T leukemic cells can use MDR1 to pump out YAP inhibitor molecules to keep the cell in a soft state, thus evading perforin pore formation.

Figure, (A) the expression of FLNA and Y357-YAP in normal human T cells and primary T leukemia cells were analyzed by western blot (n=10). (B) the expression of MDR1 in human naïve and effector T cells and primary T leukemia cells were analyzed by western blot (n=10). (C) The levels of verteporfin in CD8+ T eff cells or primary T leukemia cells from patients were analyzed by high performance liquid chromatography (HPLC). All error bars are mean \pm SD, p values were calculated by one-way ANOVA Bonferroni's test (A) or two-tailed unpaired Student's t test (B, C), n.s., not significant for $p > 0.05$ *** $p < 0.001$.

According to the reviewer's comment, we added these new results in the revised manuscript, page 12 line 252, page 11 line 228-234, Fig S7 f,g and S8a.

2. Due to the multitude of effects of the YAP-TEAD system on cancer cells, I am not sure if the currently presented data allows to conclude that the enhanced (although to a modest level) antitumor effects of CD1a CAR-T cells in the presence of YAP inhibition are indeed mediated by changes in the cell softness.

Response:

In the revised manuscript, we repeated the original CAR-T experiment using a lower effector-to-target ratio (CD1a CAR-T cell/ YAP1 KO Molt4-luc cell = 1:3). The result showed that the SGCTRL cells were weakly destroyed; in contrast, YAP1 KO cells were highly destroyed by CAR-T cells, concomitant with prolonged survival of the NSG mice (Please see figure below). However, this sensitivity of YAP1 KO cells was attenuated by treatment with latrunculin A (an inhibitor of polymerization of actin filaments, thus softening cells).

According to the reviewer's comment, we added these new results in the revised manuscript, page 10 line 216-219, fig 6a-c and S7a-c.

Moreover, we adoptively co-transferred T-leukemic cells isolated from patients and CD8+ T cells isolated from healthy donors into irradiated NSG mice. We found that the T-leukemic cells were minimally affected by allogeneic T cells; however, YAP knockout conferred the sensitivity of T-leukemic cells to allogenic T cells. Notably, such YAP1 knockout-promoted cytotoxicity was abrogated by latrunculin A or YAP/FLAN dual KO. According to the reviewer's comment, we added these new results in the revised manuscript, page 12 line 265-268, fig S8g-h.

Figure, (A-B) NSG mice were transplanted with 3×10^6 YAP knockout and vector control Molt4-luc cells. Seven days after transplantation, NSG mice were adoptively transferred with 1×10^6 human CD1a-CAR-T cells. Thirty days later, leukemia burden (spleen and bone marrow) was analyzed by flow cytometry. (C) the same as (A-B), the survival was analyzed. (D-E) the same as (A-B), except that NSG mice were treated with PBS or Lat-A. (G-H) NSG mice were irradiated (2 Gy), then adoptively transferred with 1×10^6 PHK26-labeled primary leukemic T cells (transfected with vector, YAP1 KO and YAP1/FLNA KO) and 1×10^6 allogeneic T cells. One day later, NSG mice were treated with PBS or Lat-A (20 ug/kg). On day 3, PKH26+ cells were analyzed by flow cytometry. Representative flow plots and quantification of percent of PKH26+ primary T leukemia cells in bone marrow from mice. All error bars are mean \pm SD, n.s., not significant for $p > 0.05$, ** $P < 0.01$, *** $P < 0.001$, p values were calculated by one-way ANOVA Bonferroni's test (B, E, G, H) or Log-rank test (C, F), *** $p < 0.001$.

Minor comments:

3. The current figures provide a very significant amount of non-essential confirmatory data (such as ability of perforin to permeabilize cancer cells), which dilute the novel aspects of the paper and reduce its appeal. I also suggest to separate the human and

mouse aspects of the paper, which are currently often presented in the same figures and display panels, reducing the clarity of the presentation.

Response:

We thank the reviewer's comment. We carefully checked the manuscript and removed non-essential confirmatory data, including original Fig 1, Fig S6. Moreover, we carefully separate the human and mouse data in Fig 1, Fig 2 and Fig3. These modifications should improve the clarity and highlight the novel aspects of the manuscript.

4. The abstract and introduction could benefit from additional discussion of the specific novel questions, compared to the previously reported conclusions. The paper could also benefit from the additional discussion of how the results can help to modify the current clinical testing of YAP-TEAD inhibitors.

Response:

We thank the reviewer's pertinent comment. We added the description "Our findings provide new insights into the pathway by which CTLs evades autolysis at immune synapse." In the revised Introduction (page 3, line 63).

We also added the description "In addition, our findings may be useful to modify the current clinical testing of YAP-TEAD inhibitors, considering TEAD inhibitors are currently being tested in Phase I clinical trials (Trends Biochem Sci. 2023 May;48(5):450-462)." in the revised Discussion (page 16, line 354-355).

REVIEWER COMMENTS

Reviewer #1 (Remarks to the Author):

The authors' responses to the three reviewers' critiques were thorough and appropriate. The new data they provide strengthen an already impressive body of work. This reviewer has no concerns.

Reviewer #2 (Remarks to the Author):

The authors' revisions, while improving the manuscript substantially, have not adequately addressed all of my concerns. I have listed them below.

1) The application of jasplakinolide in vivo remains poorly controlled. The experiment included in the rebuttal (top of p6) is not helpful. When were the CTLs and cancer cells removed from the mice for AFM, relative to the experimental schematic shown in Figure 5a? 24 hours after jasplakinolide injection (i.e. day 8), or 48 hours after that (i.e. day 10)? I doubt that jasplakinolide will persist in the animal at high concentrations for much more than 24 hours, so it seems likely that the cells were no longer being affected by jasplakinolide when they were extracted. Furthermore, because jasplakinolide is a reversible agent, its effects would presumably begin to diminish as soon as cells were extracted into tissue culture medium lacking the compound. Given that AFM measurements take tens of minutes to hours, it is highly unlikely that the AFM data presented actually reflect the effects of jasplakinolide on the cells in question. Accordingly, the degree to which jasplakinolide has non-cancer cell intrinsic effects in vivo, and the extent to which those effects influence cancer cell outgrowth and mouse survival, remain unresolved.

The authors argue that T cells should be resistant to jasplakinolide because they express high levels of MDR1. This runs counter to prior studies in which jasplakinolide was used to freeze the T cell cytoskeleton in vitro quite effectively (e.g. Comrie et al., JCB 208:475). Furthermore, there are indications that cyclodepsipeptides like jasplakinolide may not be subject to efflux by the MDR1 system (Zampella et al., Org. Biomol. Chem. 7:4037). The authors should at least test this hypothesis in vitro using their MDR1 inhibitor (Tariquidar).

The caveats surrounding the in vivo use of jasplakinolide also apply to the new experiments with latrunculin shown in Figure S7 and S8. Indeed, the in vivo application of latrunculin in this study is even more fraught because a loss-of-function phenotype is expected. The authors argue that the reduced therapeutic activity they observe in the presence of latrunculin is caused by cancer cell softening, but they do not (and perhaps cannot) rule out potential effects of latrunculin on T cell migration, activation, and killing. Indeed, it is well established that latrunculin treatment attenuates TCR signaling.

2) Regarding the CRISPRa screen schematized in Figure 2a, it is stated in the rebuttal that perforin treatment of control Jurkat cells renders ~10% of the cells PI+. This is not much different from the schematic FACS plot shown in Figure 2a, which I'm assuming came from a screening experiment with CRISPRa treated cells. As such, I imagine that the raw data from this screen was quite noisy, almost unmanageably so. I can't argue with the results; FLNA seems like a legitimate hit. Nevertheless, given their screen design, the authors were very lucky indeed to get it.

3) In their rebuttal, the authors justify the persistence of PI+ cancer cells in vivo by stating, with references, that macrophages and dendritic cells from NSG mice have impaired phagocytosis. I examined the two references and could find no instance in which phagocytic activity was measured directly.

4) The authors have not addressed the discrepancy between their results (which show decreased F-actin in Teff cells relative to Tnaive) and the Thauland et al paper, which shows that Teff cells contain 30% more F-actin than Tnaive (Fig S3A of that paper). The Govendir et al. paper cited by the authors is irrelevant to this issue because it measures intracellular F-actin distribution, rather than total F-actin levels.

5) It should be noted in the legend for Figure 6 that the T cells shown in panel j were restimulated ex vivo with PMA.

6) I appreciate the authors clarification regarding the use of the PKH26 dye in Figure 7. Cancer cells divide rapidly, and I am concerned that, over three days in vitro, this cell division would dilute the PKH26 to the point where the cells become invisible. Perhaps the authors could demonstrate that this is not the case in their hands?

7) Finally, I thank the authors for clarifying their perforin AFM approach, which is clearer now and easier to understand. It seems strange, however, that the perforin-induced pores shown in Figure 6e are so much smaller than the gigantic structures seen in other figures, for example Figure 2h-i. Can the authors speculate as to a reason for this discrepancy?

Reviewer #3 (Remarks to the Author):

The authors have convincingly addressed all my concerns.

RESPONSES TO REVIEWERS

We would like to express our sincere thanks to all three reviewers for their critical and constructive comments. We have performed substantial additional experiments to address Reviewer #2's concerns. We respond point-by-point to his/her comments and criticisms. We feel that his/her comments have helped us on significantly improving and strengthening the manuscript and clarifying some issues. We hope that the revision has addressed his/her major concerns.

Reviewer #2

1) The application of jasplakinolide in vivo remains poorly controlled. The experiment included in the rebuttal (top of p6) is not helpful. When were the CTLs and cancer cells removed from the mice for AFM, relative to the experimental schematic shown in Figure 5a? 24 hours after jasplakinolide injection (i.e. day 8), or 48 hours after that (i.e. day 10)? I doubt that jasplakinolide will persist in the animal at high concentrations for much more than 24 hours, so it seems likely that the cells were no longer being affected by jasplakinolide when they were extracted. Furthermore, because jasplakinolide is a reversible agent, its effects would presumably begin to diminish as soon as cells were extracted into tissue culture medium lacking the compound. Given that AFM measurements take tens of minutes to hours, it is highly unlikely that the AFM data presented actually reflect the effects of jasplakinolide on the cells in question. Accordingly, the degree to which jasplakinolide has non-cancer cell intrinsic effects in vivo, and the extent to which those effects influence cancer cell outgrowth and mouse survival, remain unresolved.

Response:

For the *in vivo* experiments, we inoculated leukemic T cells into mice for 7 days, followed by Jas treatment (i.p. injection with 50 ug/kg dose). 12 hours later, we adoptively transferred OT-1 T cells into the mice. 48 hours later, we isolated peripheral leukemic cells for AFM detection. Consistent results were obtained from 6 mice, showing that leukemic cells from the Jas-treated group was stiffer than the cells from the control group (see the original Figure 5a).

The purpose that we measured the cell stiffness by AFM was to support CTLs' killing (*Cancer Res.* 2021;81:476-88), rather than reflecting the effect of Jas. The persisting effect of Jas may be due to its indirect effect. In a support, in the revised manuscript, we used Jas to treat the cells for 24 hours and refreshed the culture medium in the absence of Jas. Either 24 or 48 hours later, the tumor cells were still stiffer than the control cells (please see the Figure below).

In addition, we found that the 24-hour Jas pretreatment had minor effect on transferred T cells. We found that 24 hours after adoptive transfer, the TNF- α ,

IL-2 and IFN- γ production and the expression of CD107a (marker for the release of perforin and granzymes) in the T cells were not affected relative to control T cells, when isolated upon *in vitro* anti-CD3/CD28 stimulation (Please see the Figure below).

The use of Jas to do the *in vivo* experiment was a proof of concept to explore new T-leukemia immunotherapy, which is significant for the clinic. The detailed exploration of Jas effect on cancer cells and non-cancer cells was beyond of the topic of this study.

A, human leukemic cells were treated with Jas for 0 or 24 hours. After Jas treatment or withdrawal, cell stiffness was determined by AFM. B-F, NSG mice were transplanted with 2×10^5 OVA-expressing ICN1 leukemic cells. On day 7 post engraftment, NSG mice were treated with Jas (50 μ g/kg), followed by adoptive transfer of 5×10^6 OT-I T cells. 24 hours later, OT-1 T cells were isolated and stimulated with anti-CD3/CD28(B). The expression of TNF- α (C), IL-2 (D), IFN- γ (E) and CD107a(F) were analyzed by flow cytometry (n=3).

The authors argue that T cells should be resistant to jasplakinolide because they express high levels of MDR1. This runs counter to prior studies in which jasplakinolide was used to freeze the T cell cytoskeleton in vitro quite effectively (e.g. Comrie et al., JCB 208:475). Furthermore, there are indications that cyclodepsipeptides like jasplakinolide may not be subject to efflux by the MDR1 system (Zampella et al., Org. Biomol. Chem. 7:4037). The authors should at least test this hypothesis in vitro using their MDR1 inhibitor (Tariquidar).

Response:

In the study by Comrie et al., Jas was used to treat CD4⁺ T cells; in contrast, we used Jas to treat CD8⁺ T cells. CD4⁺ T and CD8⁺ T cells are two different cell types and may respond to drugs differently. In addition, the Comrie's study was an *in vitro* assay without indicating concentration and time. In our study, the Jas, related to MDR1, was used for *in vivo* experiments.

In the study by Zampella, it was hard to conclude that jasplakinolide may not be subject to efflux by the MDR1 system, because (1) nine molecules, homophymines B-E and A1-E1, rather than Jas, were studied; (2) the authors mainly focused on chemical aspects and only performed very simple *in vitro* tumor cell treatment assay. Therefore, the conclusion on MDR1 was not convincing. In contrast, MDR1 has been clearly reported to have the ability to pump out peptides (*Int J Pharm.* 2007;332(1-2):147-52. *Int J Mol Sci.* 2020;22(1):246). Furthermore, high performance liquid chromatography (HPLC) analysis showed that Jas or Lat-A was mainly present in T-leukemic cells, but not in CTLs (Please see the Figure below). We did not add this result in the revised manuscript, considering our tremendous data, limited space, and the logic of writing.

Based on the above explanation and that the MDR1 inhibitor Tariquidar has been used in the YAP study in the original manuscript, we did not further repeat the MDR1 inhibition assay.

A-B, The levels of Jas (A) or Lat-A (B) in human CD8⁺ Teff cells or primary T leukemia cells(TL) from patients were analyzed by high performance liquid chromatography (HPLC).

The caveats surrounding the in vivo use of jasplakinolide also apply to the new experiments with latrunculin shown in Figure S7 and S8. Indeed, the in vivo application of latrunculin in this study is even more fraught because a loss-of-function phenotype is expected. The authors argue that the reduced therapeutic activity they observe in the presence of latrunculin is caused by cancer cell softening, but they do not (and perhaps cannot) rule out potential effects of latrunculin on T cell migration, activation, and killing. Indeed, it is well established that latrunculin treatment attenuates TCR signaling.

Response:

In our previous study, we used blebbistatin (Ble), an inhibitor of myosin II ATPase activity, to pretreated B16 melanoma-bearing mice, followed by CD8⁺ tumor-specific T cell adoptive transfer. We observed Bleb inhibited CTLs killing of B16 tumor cells (*Cancer Res.* 2021;81(2):476-488).

In the present study, we used the actin-depolymerizing macrolide latrunculin A (Lat-A) to pretreat T-leukemia-bearing mice, followed by CD45.1⁺ OT-I cells adoptive transfer 24 hour later. This 24-hour pretreatment may effectively soften tumor cells but have minor effect on transferred T cells. In our unpublished data, we actually analyzed the transferred T cell function. We found that 24 hours after adoptive transfer, the TNF- α , IL-2 and IFN- γ production and expression of CD107a (marker for the release of perforin and granzymes) in the T cells were not affected relative to control T cells, when isolated upon *in vitro* anti-CD3/CD28 stimulation (Please see the Figure below). We did not add this in the manuscript, considering our tremendous data and limited space and concise content.

A-D, NSG mice were transplanted with 2×10^5 OVA-expressing ICN1 leukemic cells. At 7 days post engraftment, NSG mice were treated or untreated with Lat-A (20 μ g/kg), followed by adoptively transferring with or without 5×10^6 OT-I T cells. At 24-hour post engraftment, CD45.1 OT-1 were isolated by flow cytometry. Results from flow cytometric analysis of the expression of TNF- α (A), IL-2 (B), IFN- γ (C) and CD107a(D) were shown (n=3).

2) Regarding the CRISPRa screen schematized in Figure 2a, it is stated in the rebuttal that perforin treatment of control Jurkat cells renders ~10% of the cells PI+. This is not much different from the schematic FACS plot shown in Figure 2a, which I'm assuming came from a screening experiment with CRISPRa treated cells. As such, I imagine that the raw data from this screen was quite noisy, almost unmanageably so. I can't argue with the results; FLNA seems like

a legitimate hit. Nevertheless, given their screen design, the authors were very lucky indeed to get it.

Response:

In our previous rebuttal, we described "When we used 50 ng/ml of perforin to treat Jurkat cells, we found that more than 90% of the cells were resistant (negative for PI staining). Intriguingly, introducing a CRISPRa gRNA library into Jurkat cells substantially increased their sensitivity to perforin (positive for PI staining)". In the presence of perforin, approximately 20-30% of Jurkat cells with the CRISPRa gRNA library exhibited positive PI staining, compared to only 3-5% in untreated control cells. To ensure the specificity of screening, we only sorted the top 10% of cells with positive PI staining and the bottom 10% with negative PI staining for next-generation sequencing (NGS). Our data, verified by two NGS biological replicates and a series of experiments, demonstrated that FLNA mediates the sensitivity of T-cells and T-leukemia cells to perforin. We thank the reviewer's indicating this issue, and in the revised manuscript, we modified the schematic FACS plot to clearly show the experiment process (Please see the Figure below).

A, Jurkat with or without CRISPRa gRNA library were treated with perforin for 10 min. The PI⁺ cells were analyzed by flow cytometry.

3) In their rebuttal, the authors justify the persistence of PI⁺ cancer cells in vivo by stating, with references, that macrophages and dendritic cells from NSG mice have impaired phagocytosis. I examined the two references and could find no instance in which phagocytic activity was measured directly.

Response:

Tumor cell death is a common phenomenon in the host, due to hypoxia, nutrient deprivation, radiation, chemotherapy and/or immuno-killing. Treatments, including T cell immunotherapy, may result in continuously tumor cell death, which can be isolated for further study. And this ex vivo analysis of dead tumor cells has been widely conducted in cancer research laboratories, including ours. A common observation is that when tumor tissues are digested with enzymes into single cells, many dead tumor cells are present in the bulk population, which should be excluded during flow cytometric analysis (*STAR Protoc.* 2021;2(4):100841; *STAR Protoc.* 2023;4(1):101951). As the reason why the PI⁺

cells were not rapidly cleared, we think that it is beyond the scope of this study.

The two references indicated that NSG mice poorly produce Th1 cytokines, which commonly stimulate macrophage polarization toward M1. And M1 macrophages have a stronger phagocytic ability than M2 macrophages (*Cell Discov.* 2021;7(1):24); however, tumors commonly educate macrophages into M2. Therefore, we cited these two references. The definite indication of that NSG mice have impaired phagocytosis can be found from the report (*Biol Trace Elem Res.* 2018;184(1):196-205).

4) *The authors have not addressed the discrepancy between their results (which show decreased F-actin in Teff cells relative to Tnaive) and the Thauland et al paper, which shows that Teff cells contain 30% more F-actin than Tnaive (Fig S3A of that paper). The Govendir et al. paper cited by the authors is irrelevant to this issue because it measures intracellular F-actin distribution, rather than total F-actin levels.*

Response:

The discrepancy might be explained by two reasons:

The first was that Thauland et al. used flow cytometry to analyze the average fluorescence intensity of F-actin in fixed CD4⁺ Teff cells. In contrast, we used immunofluorescence microscope to observe F-actin in live CD8⁺ T cells. CD4⁺ and CD8⁺ T cells are different cell types. Also, the method used was also different.

The second was that effector T cells are strikingly larger than naïve T cells. Therefore, even the total amount of F-actin is larger in Teff cells than in naïve T cells, the F-actin per unit volume may be less in Teff than in naïve T cells.

5) *It should be noted in the legend for Figure 6 that the T cells shown in panel j were restimulated ex vivo with PMA.*

Response:

In the revised manuscript, we modified the description of the legend for fig 6J as below:

"T cells were stimulated with PMA and treated with 0.1µM VP for 24 hours. Flow cytometric analysis was performed to evaluate the expression of IFN-γ and TNFα."

6) *I appreciate the authors clarification regarding the use of the PKH26 dye in Figure 7. Cancer cells divide rapidly, and I am concerned that, over three days in vitro, this cell division would dilute the PKH26 to the point where the cells*

become invisible. Perhaps the authors could demonstrate that this is not the case in their hands?

Response:

PKH26 is a red fluorescent dye generally used for cell membrane labeling. It has been characterized in a number of model systems and widely used in *in vivo* cell tracking. In an early study, PKH26-labeled B leukemic cells could be detected for up to 14 days and maintained a high level of PKH26 positivity within the first 6 days (*Leuk Res.* 1995;19(2):113-20). In our experiments, after labeling leukemia cells with PKH26, we continuously tracked them *in vitro* for 3 days. Through flow cytometry, we observed that PKH26 fluorescence intensity decreased as cells underwent division, but the fluorescence signal remained detectable throughout the tracking period (Please see the Figure below).

A

A, NSG mice were irradiated (2 Gy), then adoptively transferred with 1×10^6 PKH26-labeled primary leukemic T cells and 1×10^6 allogeneic T cells. Results from flow cytometric analysis of the expression of PKH26 are shown.

7) Finally, I thank the authors for clarifying their perforin AFM approach, which is clearer now and easier to understand. It seems strange, however, that the perforin-induced pores shown in Figure 6e are so much smaller than the gigantic structures seen in other figures, for example Figure 2h-i. Can the authors speculate as to a reason for this discrepancy?

Response:

The reason might be attributed to the different perforin quantity. The *ex vivo* (Fig. 6e) perforin forming pores was not as strong as *in vitro* (Fig. 2h-i), considering the use of high dose of perforin (50 ng/ml) *in vitro*. Our previous studies have found that higher concentration of perforin can form larger pores (*Cell Mol Immunol.* 2019;16(6):611-620.).

REVIEWER COMMENTS

Reviewer #2 (Remarks to the Author):

The authors' most recent response has done little to assuage my concerns about the Jasplakinolide and Latrunculin experiments.

1) The authors' description of their experimental protocol leaves little doubt in my mind that no actual Jasplakinolide remained in any of the leukemic cells when they were analyzed ex vivo by AFM. That being the case, the apparent stiffening of these cells must have been some sort of secondary effect, a conclusion that the authors acknowledge in their response. If the authors intend to proceed with this rationale, it is something that should be covered in their Discussion. In other words, they should write a paragraph about how they think Jasplakinolide continues to stiffen cells after being washed away. Part A of the first figure in their response should also be included in the actual manuscript.

2) The Latrunculin A experiments remain troubling. In their response, the authors suggest that the Latrunculin selectively affected cancer cells, rather than CTLs, in their in vivo experiments because the Latrunculin was injected 24 h before the T cells. Hence, the drug would have been mostly excreted prior to T cell arrival. This rationale implies that they believe that Latrunculin, like Jasplakinolide, also has "memory" effects, in that cancer cells treated with Latrunculin remain softer even after the drug has been removed. If they would like to make this argument, they should provide some evidence that Latrunculin actually affects cells in this way.

More disturbingly, the description of their Latrunculin experiments in the response document does not match the methods in their manuscript.

From the response document:

In the present study, we used the actin-depolymerizing macrolide latrunculin A (Lat-A) to pretreat T-leukemia-bearing mice, followed by CD45.1+ OT-I cells adoptive transfer 24 hour later. This 24-hour pretreatment may effectively soften tumor cells but have minor effect on transferred T cells.

(Latrunculin was delivered first, enabling the authors to argue that the drug was partially or completely excreted by the time the T cells were injected)

However, from the legend for Supp Figure 7:

a) NSG mice were transplanted with 3×10^6 YAP knockout and vector control Molt4 cells. At 7 days post engraftment, NSG mice were adoptively transferred with 1×10^6 human CD1a-CAR-T cells, at 30 days post engraftment, NSG mice treated or untreated with latrunculin A (20 ug/kg) and leukemia burden was measured by flow cytometry.

(T cells are applied first (day 7) followed by Latrunculin (day 30). Also, CAR T cells, and not OT-1 T cells, were used.)

Legend to Supp Figure 8:

g, h NSG mice were irradiated (2 Gy), then adoptively transferred with 1×10^6 PHK26-labeled primary leukemic T cells transfected with vector, YAP1 KO or YAP1/FLNA dual KO and 1×10^6 allogeneic T cells. One day later, NSG mice were treated with PBS or Lat-A (20 ug/kg).

(Again, T cells were added BEFORE Latrunculin).

Either latrunculin was injected before the T cells or it wasn't. Either way, the authors will need to do more to justify their interpretations and conclusions. But first, they have to figure out what protocol was actually used.

3) I remain unconvinced about the relevance of MDR1 to the apparent differential sensitivity of T cells and leukemic cells to Jaspilakinolide and Latrunculin. Why can't the authors just measure the effects of Jaspilakinolide and Latrunculin in the presence of Tariquidar? This seems like a fairly straightforward experiment.

Also, their figure showing higher levels of Jaspilakinolide and Latrunculin in leukemic cells relative to T cells is not compelling. These are 2-3 fold differences, which means little to nothing in pharmacological experiments. As an example, would one really expect 2-3 uM Jaspilakinolide to induce dramatically different effects than 1 uM Jaspilakinolide? I'm trying to remember an instance when a 2.5 fold difference in the concentration of a small molecular inhibitor actually made a difference in a biological experiment . . .

RESPONSES TO REVIEWERS

Reviewer #2 (Remarks to the Author):

The authors' most recent response has done little to assuage my concerns about the Jasplakinolide and Latrunculin experiments.

1) The authors' description of their experimental protocol leaves little doubt in my mind that no actual Jasplakinolide remained in any of the leukemic cells when they were analyzed ex vivo by AFM. That being the case, the apparent stiffening of these cells must have been some sort of secondary effect, a conclusion that the authors acknowledge in their response. If the authors intend to proceed with this rationale, it is something that should be covered in their Discussion. In other words, they should write a paragraph about how they think Jasplakinolide continues to stiffen cells after being washed away. Part A of the first figure in their response should also be included in the actual manuscript.

Response:

In this study, we found that tyrosine phosphorylation of YAP physiologically softens effector T cells, leading to the cells evading perforin pore formation at the immune synapse. At the same time, we also found that T leukemic cells, a very malignant leukemia type that are usually resistant to T cell killing, are intrinsically soft, prompting us to test a hypothesis that selectively stiffening T leukemic cells may strengthen T cell killing of them.

Cell stiffness/softness is controlled by cytoskeletons, especially by microfilaments. However, there are no ideal agents available to target F-actin (both Jas and Lat-A are toxic). We only used Jas as a tool to enhance the leukemic cell stiffness in the experiments. In the original Figure 5a, we transferred leukemic T cells into mice, followed by Jas treatment (i.p. injection with 50 ug/kg dose). 12 hours later, we adoptively transferred OT-1 T cells into the mice. 48 hours later, we isolated peripheral leukemic cells for AFM detection. We found that leukemic cells from the Jas-treated group was stiffer than the cells from the control group. This result was consistent with our *in vitro* experiment, which was shown in the last round point-by-point reply.

We understand the reviewer's curiosity. However, the reason why injection of Jas once allowed T leukemic cells to maintain a stiff state 48 hour later was really beyond the focus of this study. The possibility may be a stable binding of Jas to F-actin or Jas-caused damage makes T leukemic cells difficult to recover the normal F-actin homeostasis in a short time. We did not add this information in the revised manuscript, considering the crowded data and limited space and the logic of the study.

For the reviewer' mentioned part A of the displayed Figure, we added it to the

2) The Latrunculin A experiments remain troubling. In their response, the authors suggest that the Latrunculin selectively affected cancer cells, rather than CTLs, in their *in vivo* experiments because the Latrunculin was injected 24 h before the T cells. Hence, the drug would have been mostly excreted prior to T cell arrival. This rationale implies that they believe that Latrunculin, like Jasplakinolide, also has “memory” effects, in that cancer cells treated with Latrunculin remain softer even after the drug has been removed. If they would like to make this argument, they should provide some evidence that Latrunculin actually affects cells in this way.

Response:

Like Jas, Lat-A also binds to F-actin, where it, however, depolymerizes actin filaments, thereby softening cells. In the revised manuscript, we used Lat-A to treat YAP1-KO T leukemic cells for 24 hours and refreshed the culture medium in the absence of Lat-A. 48 hours later, the tumor cells were still softer than the control cells (please see the Figure below).

A, YAP1-KO MOLT-4 cells were treated without or with Lat-A (0.1 μ M) for 24 hours. After Lat-A treatment or withdrawal, cell stiffness was determined by AFM.

This Lat-A-resulted phenomenon was similar to Jas treatment. The reviewer called this “memory” effect, which might also be stemmed from the stable binding of Lat-A to F-actin or Lat-A-caused damage that makes tumor cells difficult to restore the homeostasis of normal F-actin in a short time.

More disturbingly, the description of their Latrunculin experiments in the response document does not match the methods in their manuscript.

From the response document:

In the present study, we used the actin-depolymerizing macrolide latrunculin A (Lat-A) to pretreat T-leukemia-bearing mice, followed by CD45.1+ OT-I cells adoptive transfer 24 hour later. This 24-hour pretreatment may effectively soften tumor cells but have minor effect on transferred T cells.

(Latrunculin was delivered first, enabling the authors to argue that the drug was

partially or completely excreted by the time the T cells were injected)

However, from the legend for Supp Figure 7:

a) NSG mice were transplanted with 3×10^6 YAP knockout and vector control Molt4 cells. At 7 days post engraftment, NSG mice were adoptively transferred with 1×10^6 human CD1a-CAR-T cells, at 30 days post engraftment, NSG mice treated or untreated with latrunculin A (20 ug/kg) and leukemia burden was measured by flow cytometry.

(T cells are applied first (day 7) followed by Latrunculin (day 30). Also, CAR T cells, and not OT-1 T cells, were used.)

Legend to Supp Figure 8:

g, h NSG mice were irradiated (2 Gy), then adoptively transferred with 1×10^6 PHK26-labeled primary leukemic T cells transfected with vector, YAP1 KO or YAP1/FLNA dual KO and 1×10^6 allogeneic T cells. One day later, NSG mice were treated with PBS or Lat-A (20 ug/kg).

(Again, T cells were added BEFORE Latrunculin).

Either latrunculin was injected before the T cells or it wasn't. Either way, the authors will need to do more to justify their interpretations and conclusions. But first, they have to figure out what protocol was actually used.

Response:

In the first-round revision, this reviewer and another reviewer (reviewer #3) concerned whether the in vivo enhanced T cell killing by YAP inhibition is due to the changes in cell stiffness. To address this question, we introduced Lat-A as a cell softening agent to soften YAP1 KO leukemia cells in NSG mice. Our results demonstrated that Lat-A indeed impairs the killing of YAP1-KO tumor cells by OT-1, CAR-T or allogeneic T cells. Evaluating these results, Reviewer #3 stated "The authors have convincingly addressed all my concerns." However, Reviewer #2 (this reviewer) still expressed concerns about the application of Jas or Lat-A in vivo, given the possible damage of the functionality of CD8+ T cells. Therefore, in the second round of revision, we carefully detected CD8+ T cell functionality. We provided clearly data to show that CD8+ T cells are rarely affected by the agents.

As for the difference between the in vivo studies of Fig 5a and supp Figs 7 and 8, This was because these experiments were designed to address different questions with a brilliant logic. We designed experiment of Fig 5a to address the reviewer's concern, where tumor cells were pretreated with Jas or Lat-A before adoptively transferring OT-1 CD8+ T cells. This protocol was different from the protocol for Supp Fig 7a and Supp Fig 8 g,h, lying in that T cell were added after Lat-A in Fig 5a but before Lat-A in Supp Figs 7 and 8.

Following the experiments of Figure 5, we tried to use YAP inhibitor to enhance

T leukemic cell stiffness. However, this may also enhance effector cell stiffness, because the stiffness of both cell types is subject to YAP regulation, which may lead to autolysis of tumor-specific T cells during the killing process. However, we conducted experiments in Figure 6, showing that effector T cells highly express MDR1 to efficiently pump drug/inhibitor molecules out. This suggests that YAP inhibitor can be used to selectively target T leukemic cells. To imitate endogenous tumor-reactive T cells to attack T leukemic cells in patients, we administrated drugs following T cell adoptively transfer.

Therefore, following Figure 6, we further conducted experiments in Supp Figs 7a and 8g, h, which showed that CAR-T and allogeneic T cells were cytotoxic to YAP1 KO primary T leukemic cells. However, the addition of latrunculin A after T cells abrogated the cytotoxicity of the T cells. To address the reviewer's concerns, in the revised manuscript, we evaluated the possible impact of Lat-A on CAR-T and allogeneic T cells. We conducted experiments similar to those shown in Supplementary Figures 7a and 8g. The results showed that Lat-A had no effect on the stiffness of effector T cells but did soften T leukemic cells (please see the Figure A or C below). In line with this, we observed that the effector functions of CAR-T and allogeneic T cells were not affected by the administration of low dose Lat-A (please see the Figure B or D below). We added these results in the revised manuscript, page 10 line 222 or page 13 line 276, Fig. S7, f to g or Fig. S10, c to d.

A-B, NSG mice were transplanted with 3×10^6 YAP1 KO Molt4-luc cells. At 7 days post engraftment, NSG mice were adoptively transferred with 1×10^6 human CD1a-CAR, followed by treated with Lat-A(20 μ g/kg). 24 hours later, GFP⁺ CAR-T cells and Molt4-luc cells were isolated. The stiffness of CAR-T cells and YAP1 KO Molt4-luc cells was determined by AFM (A). CAR-T

stimulated with PMA and ionomycin and the expression of TNF- α , IL-2, IFN- γ and CD107a were analyzed by flow cytometry (n=6) (B). C-D, NSG mice were irradiated (2 Gy), then adoptively transferred with 1×10^6 primary leukemic T cells labeled with PHK26 and 1×10^6 allogeneic T cells labeled with CFSE. At 1 day post engraftment, NSG mice were treated with Lat-A (20 μ g/kg). At 3 days post engraftment, CFSE⁺ and PHK26⁺ cells were isolated respectively. The stiffness of allogeneic T cells and primary leukemic T cells was determined by AFM (C). Allogeneic T cells were stimulated with PMA and ionomycin. The expression of TNF- α , IL-2, IFN- γ and CD107a were analyzed by flow cytometry (n=6) (D).

In addition, the same results were obtained in the experiment of treating allogeneic T cells with low dose Jas (please see the Figure below). We did not add this result in the revised manuscript, considering the logic of writing.

A, NSG mice were irradiated (2 Gy), then adoptively transferred with 1×10^6 primary leukemic T cells labeled with PHK26 and 1×10^6 allogeneic T cells labeled with CFSE. At 1 day post engraftment, NSG mice were treated with Jas (50 μ g/kg). At 3 days post engraftment, CFSE⁺ allogeneic T cells were isolated and stimulated with PMA and ionomycin. The expression of TNF- α , IL-2, IFN- γ and CD107a were analyzed by flow cytometry (n=6).

Finally, we appreciate the reviewer's indicating an error in figure legend of Supp Fig 7a. According to the reviewer's comment, we added the description in the revised Figure legend of Fig. S7a with a modification as below:

“NSG mice were transplanted with 3×10^6 YAP knockout and vector control Molt4 cells. At 7 days post engraftment, NSG mice were adoptively transferred with 1×10^6 human CD1a-CAR-T cells, followed by treating with PBS or latrunculin A (20 ug/kg, intraperitoneally) every 3 d, for a total of 3 times. At 30 days post engraftment, leukemia burden was measured by flow cytometry. “

3) I remain unconvinced about the relevance of MDR1 to the apparent

differential sensitivity of T cells and leukemic cells to Jasplakinolide and Latrunculin. Why can't the authors just measure the effects of Jasplakinolide and Latrunculin in the presence of Tariquidar? This seems like a fairly straightforward experiment.

Also, their figure showing higher levels of Jasplakinolide and Latrunculin in leukemic cells relative to T cells is not compelling. These are 2-3 fold differences, which means little to nothing in pharmacological experiments. As an example, would one really expect 2-3 uM Jasplakinolide to induce dramatically different effects than 1 uM Jasplakinolide? I'm trying to remember an instance when a 2.5 fold difference in the concentration of a small molecular inhibitor actually made a difference in a biological experiment . . .

Response:

In the revised manuscript, we used the MDR1 inhibitor Tariquidar to directly treat T cells and T leukemic cells, respectively, in the presence of Jas. The result showed that the use of an MDR1 inhibitor increased Jas contents in CD8+ T cells, which also resulted in stiffening CD8 T cells and increasing the number of perforin-formed pores (please see the Figure A-D below). Additionally, when we transfected T leukemia cells with pCMV-MDR1 plasmid, we found that intracellular Jas decreased in T leukemic cells. In line with this, transfected T leukemia cells resisted to Jas-induced stiffness and evaded perforin-mediated pore formation (please see the Figure E-H below). We added it to the revised manuscript, page 11 line 238 and fig S8 c-j.

A-B, human CD8+ T cells were treated with 200nM Jas combined with or without 1nM Tariquidar for 12 hr. The levels of Jas in human CD8+ Teff cells were analyzed by high performance liquid chromatography (HPLC) (A). The stiffness of cell was analyzed by AFM (B). C-D, the same as (B), except that human CD8+ T cells were treated with Perforin for 10 min then were analyzed by flow cytometry (C) or AFM (D). The formed pore size and number were calculated. E-F, the same as (A-B), except that human CD8+ T cells transfected with or without pCMV-MDR1 were treated with 200nM Jas for 12 hr. G-H, the same as (C-D), except that human CD8+ T cells was transfected with or without pCMV-MDR1.

Similar result was obtained in Tariquidar-treated YAP-KO cells in the presence of Lat-A (please see the Figure below). We did not add this in the manuscript, considering our tremendous data and limited space and the logic of writing.

A-B, human YAP1 KO CD8+ T cells were treated with 100nM Lat-A combined with or without 1nM Tariquidar for 12 hr. The levels of Lat-A in human YAP1 KO CD8+ Teff cells were analyzed by high performance liquid chromatography (HPLC) (A). The stiffness of cell was analyzed by AFM (B). C, the same as (B), except that human YAP1 KO CD8+ T cells were treated with Perforin for 10 min then were analyzed by flow cytometry (C).

For the Jas and Lat-A content, we did not agree the reviewer's comment. Both Jas and Lat-A are very toxic to cells. From the statistical aspect, 2.5-fold difference is extremely significant. We were confused with the reviewer mentioned "*would one really expect 2-3 uM Jasplakinolide to induce dramatically different effects than 1 uM Jasplakinolide*". It is not clear that 2-3uM Jas is used to treat cells or 2-3 uM Jas means intracellular concentration. If it was the former, this did not represent the cytoplasmic concentration in the cells; and if it was the latter, 2-3 uM Jas may be enough to stiffen the cells by binding to F-actin, compared to 1 uM Jas. The reviewer should not ignore the authors' results, showing that low dose Jas treatment stiffened T leukemic cells rather than effector T cells (please see panel A and C of the third reply).

In addition, it has been reported that compared to a dose of 0.05 μ M, 0.1 μ M and 0.2 μ M of Jasplakinolide significantly reduced SNTA1 tyrosine phosphorylation (Protein J. 2021 Apr;40(2):234-244., please see figure 4a in this article). Also, it has been reported that 3 μ M of Latrunculin A was unable to inhibit hypoxia-induced HIF-1 activity, whereas 5.6 μ M of Lat-A resulted in a 40% inhibition (J Nat Prod. 2008 Mar;71(3):396-402, please see figure 4 in this article).

REVIEWERS' COMMENTS

Reviewer #2 (Remarks to the Author):

I've organized my response around the specific points of contention covered in the authors' most recent rebuttal letter.

1) Sustained effects of Jasplakinolide and Latrunculin:

Jasplakinolide and Latrunculin are widely thought to be reversible agents, and they have been used in this manner in numerous studies. Accordingly, the authors' in vivo results make no sense in the absence of data substantiating the "memory" effects of both compounds. I am glad that the authors now include data showing prolonged stiffness modulation by Jasplakinolide after withdrawal (Fig. S6a). They should also include the Latrunculin withdrawal figure (the second figure in their rebuttal) in the manuscript. Furthermore, in their Discussion, the authors should draw attention to the fact that their results imply that Latrunculin and Jasplakinolide have prolonged effects on cell stiffness that persist for hours after washout. In their rebuttal, they speculated as to some potential mechanisms for this mechanical memory. This speculation should also be included in the Discussion. This is information that the field needs to know because both compounds are widely used and, as I've stated above, they are generally assumed to be reversible.

2) Confusion about protocol for Latrunculin treatment in vivo:

In their rebuttal, the authors contrast the protocol for Latrunculin treatment in Figure 5 with the protocol used in Figures S7 and S8. Figure 5, however, makes no mention of Latrunculin at all. Nor is Latrunculin referenced in the corresponding portion of the manuscript text. As such, I have no idea what the authors are talking about. They seem to be calling out an experiment that they apparently did not perform. Am I missing something here? It is difficult for me to appreciate the "brilliance" of their logic when the relevant data are absent from the manuscript.

3) Drug efflux and differential sensitivity:

The authors' new data addresses my concerns. Nice work.

REVIEWERS' COMMENTS Reviewer #2 (Remarks to the Author):

I've organized my response around the specific points of contention covered in the authors' most recent rebuttal letter.

1) Sustained effects of Jasplakinolide and Latrunculin: Jasplakinolide and Latrunculin are widely thought to be reversible agents, and they have been used in this manner in numerous studies. Accordingly, the authors' in vivo results make no sense in the absence of data substantiating the "memory" effects of both compounds. I am glad that the authors now include data showing prolonged stiffness modulation by Jasplakinolide after withdrawal (Fig. S6a). They should also include the Latrunculin withdrawal figure (the second figure in their rebuttal) in the manuscript. Furthermore, in their Discussion, the authors should draw attention to the fact that their results imply that Latrunculin and Jasplakinolide have prolonged effects on cell stiffness that persist for hours after washout. In their rebuttal, they speculated as to some potential mechanisms for this mechanical memory. This speculation should also be included in the Discussion. This is information that the field needs to know because both compounds are widely used and, as I've stated above, they are generally assumed to be reversible.

Response:

We thank the reviewer's suggestion.

According to the reviewer's comment and suggestion, we added the result of the latrunculin withdrawal in the revised manuscript, page 10 line 206 and supp fig 6b.

In the revised manuscript, we also added the discussion of the prolonged effect of latrunculin and jasplakinolide on cell stiffness observed in our study (please see page 18 line 367-371). Meanwhile, we discuss the potential mechanisms underlying this mechanical memory, as mentioned in our rebuttal.

"Similar concerns are derived from the use of cytoskeletal drugs, which stably bind to F-actin, thus resulting in cytotoxicity⁷⁰. In this study, we found that once injection of Jas may maintain a 48-hour stiffness of T leukemic cells, which might be explained by a stable binding of Jas to F-actin or that Jas-caused damage makes T leukemic cells difficult to recover the normal F-actin homeostasis within a short time. Despite these, in this study, we demonstrated that low dose YAP inhibitor is an ideal choice."

2) Confusion about protocol for Latrunculin treatment in vivo: In their rebuttal, the authors contrast the protocol for Latrunculin treatment in Figure 5 with the protocol used in Figures S7 and S8. Figure 5, however, makes no mention of Latrunculin at all. Nor is Latrunculin referenced in the corresponding portion of the manuscript text. As such, I have no idea what the authors are talking about. They seem to be calling out an experiment that they apparently did not perform. Am I missing something here? It is difficult for me to appreciate the "brilliance" of their logic when the relevant data are absent from the manuscript.

Response:

In the last revised manuscript, we conducted substantial experiments and generated clear data related to Latrunculin treatment in OT-1 mouse model to address the reviewer's concerns. As for the part mentioned here by the reviewer, we described the experimental process and the results in the last rebuttal; however, we did not include the experimental results in the revised manuscript due to the logic and main

topic of this study as well as the crowded data and limited space (a total of 9 main figures and 11 supplementary figures in the last revised manuscript).

3) Drug efflux and differential sensitivity: The authors' new data addresses my concerns. Nice work.

Response:

We thank the reviewer's comment.